Citation: *Molecular Systems Biology* 9:685
www.molecularsystemsbiology.com

# Generalized bacterial genome editing using mobile group II introns and Cre-*lox*

Peter J Enyeart[1], Steven M Chirieleison[2,5], Mai N Dao[3,4], Jiri Perutka[1,3], Erik M Quandt[1], Jun Yao[1,3,4], Jacob T Whitt[1,3,4], Adrian T Keatinge-Clay[1,3], Alan M Lambowitz[1,3,4] and Andrew D Ellington[1,3,*]

[1] Institute for Cell and Molecular Biology, University of Texas at Austin, Austin, TX, USA, [2] Department of Biomedical Engineering, University of Texas at Austin, Austin, TX, USA, [3] Department of Chemistry and Biochemistry, University of Texas at Austin, Austin, TX, USA and [4] Section of Molecular Genetics and Microbiology, School of Biological Sciences, University of Texas at Austin, Austin, TX, USA
[5] Present address: School of Medicine, Case Western Reserve University, Cleveland, OH 44106, USA.
* Corresponding author. Institute for Cell and Molecular Biology, University of Texas at Austin, Austin, TX 78712, USA. Tel.: + 1 512 232 3424; Fax: + 1 512 471 7014; E-mail: andy.ellington@mail.utexas.edu

Efficient bacterial genetic engineering approaches with broad-host applicability are rare. We combine two systems, mobile group II introns ('targetrons') and Cre/*lox*, which function efficiently in many different organisms, into a versatile platform we call GETR (Genome Editing via Targetrons and Recombinases). The introns deliver *lox* sites to specific genomic loci, enabling genomic manipulations. Efficiency is enhanced by adding flexibility to the RNA hairpins formed by the *lox* sites. We use the system for insertions, deletions, inversions, and one-step cut-and-paste operations. We demonstrate insertion of a 12-kb polyketide synthase operon into the *lacZ* gene of *Escherichia coli*, multiple simultaneous and sequential deletions of up to 120 kb in *E. coli* and *Staphylococcus aureus*, inversions of up to 1.2 Mb in *E. coli* and *Bacillus subtilis*, and one-step cut-and-pastes for translocating 120 kb of genomic sequence to a site 1.5 Mb away. We also demonstrate the simultaneous delivery of *lox* sites into multiple loci in the *Shewanella oneidensis* genome. No selectable markers need to be placed in the genome, and the efficiency of Cre-mediated manipulations typically approaches 100%.
*Molecular Systems Biology* 9: 685; published online 3 September 2013; doi:10.1038/msb.2013.41
*Subject Categories:* synthetic biology; genome stability & dynamics
*Keywords:* bacterial genome engineering; Cre-*lox*; mobile group II introns; *Staphylococcus aureus*; *Shewanella oneidensis*

## Introduction

Though synthetic biology has thus far been focused primarily on building circuits of small numbers of genes to perform tasks of interest (Kaern *et al*, 2003; Lu *et al*, 2009), in recent years, more interest is being taken in the genome as a whole as the unit of engineering (Gibson *et al*, 2010; Dymond *et al*, 2011; Isaacs *et al*, 2011). As interest in engineering bacterial genomes increases, the need for efficient tools for manipulating these genomes will also increase. Though a variety of methods exist for engineering bacterial genomes (Miller, 1991; Hughes and Maloy, 2007), each has specific limitations in terms of site specificity, efficiency, versatility, and/or range of applicable bacterial species. Recombineering and related methods making use of phage recombinases have come into widespread use for small-scale modifications in *Escherichia coli* (Datsenko and Wanner, 2000; Yu *et al*, 2000; Costantino and Court, 2003; Wang *et al*, 2009), but use of this approach in other species has so far been limited and often requires developing new recombineering functions for each system (Datta *et al*, 2008; van Kessel and Hatfull, 2008; Swingle *et al*, 2010). On the other hand, site-specific recombinases such as the Cre-*lox* system are quite efficient and function in many organisms; indeed, the Cre-*lox* system has been claimed to function efficiently 'in any cellular environment and on any kind of DNA' (Nagy, 2000). In bacteria, the system has thus far been primarily used for selective marker removal, but it has, for example, been used to create large deletions in *E. coli* (Fukiya *et al*, 2004) and large inversions in *Lactococcus lactis* (Campo *et al*, 2004). However, positioning the recombination-recognition targets requires complementary genome-engineering approaches (typically with selectable markers), thus creating a chicken-and-egg problem.

Retargetable mobile group II introns are an another tool that has been developed relatively recently. These so-called 'targetrons' can be designed to insert into a given DNA site at efficiencies high enough that selectable markers need not be used. Mobile group II introns occur naturally in bacteria, eukaryotic organelles, and some archaea, and are thought to be precursors to the eukaryotic spliceosome (Lambowitz and Zimmerly, 2004). In these introns, the intron-encoded protein

(IEP) aids in self-splicing and in the process of 'retrohoming,' in which the intron site-specifically reverse splices into DNA. Diagrams of the intron structure and the mechanism of intron retrohoming are shown in Figure 1. Retrohoming sites are recognized primarily by base-pairing interactions between the intron RNA and target DNA, and it is therefore possible to change the specificity of intron insertion by modifying the target-site recognition sequences in the intron RNA. Algorithms have been developed for efficiently retargeting both the Ll.LtrB intron from *L. lactis* (Perutka *et al*, 2004) and the EcI5 intron from *E. coli* (Zhuang *et al*, 2009) and are available online (www.targetrons.com). The two intron types have little apparent sequence homology.

Targetrons function in a wide variety of bacteria. Rodriguez *et al* (2009) compiled a list of 11 different bacterial genera in which targetrons have been shown to function, and at least 9 more genera have been added to the list of known targets since that time (Alonzo *et al*, 2009; Zarschler *et al*, 2009; Park *et al*, 2010; Steen *et al*, 2010; Kumar *et al*, 2011; Palonen *et al*, 2011; Akhtar and Khan, 2012; Cheng *et al*, 2013; Smith *et al*, 2013). For some genera, such as *Clostridia*, targetrons have proven to be the first genetic tool of significant utility beyond suicide plasmids, which have low efficiency and are unstable (Heap *et al*, 2007).

While targetrons are conventionally used for gene knock-outs, their efficiency, specificity, and broad applicability make them attractive for tandem use with other general-utility genome-engineering tools, such as site-specific recombinases. Here, we modified targetrons to efficiently carry *lox* sites to defined genomic loci and thereby developed a generalizable approach to genome editing that can be adapted with minimal modification to a wide variety of bacterial strains. We use this system, called GETR (Genome Editing via Targetrons and Recombinases), to generate large-scale chromosomal insertions, deletions, inversions, and one-step cut-and-pastes, and we demonstrate its use in the Gram-negative *E. coli* and *Shewanella oneidensis* bacteria, as well as in the Gram-positive *Staphylococcus aureus* and *Bacillus subtilis* bacteria.

## Results

### Engineering *lox*-site inserts for improved intron efficiency

One of the most well-understood site-specific recombinases is the Cre/*lox* system, first discovered in 1981 in the

P1 bacteriophage (Sternberg and Hamilton, 1981). The Cre protein catalyzes recombination between *lox* sites. *Lox* sites are 34 nucleotides long and consist of 13-nucleotide palindromic repeats flanking an 8-nucleotide linker (Hoess and Abremski, 1984). The linkers are asymmetrical and thus have a specific orientation that controls the directionality of recombination by the Cre protein (see Supplementary

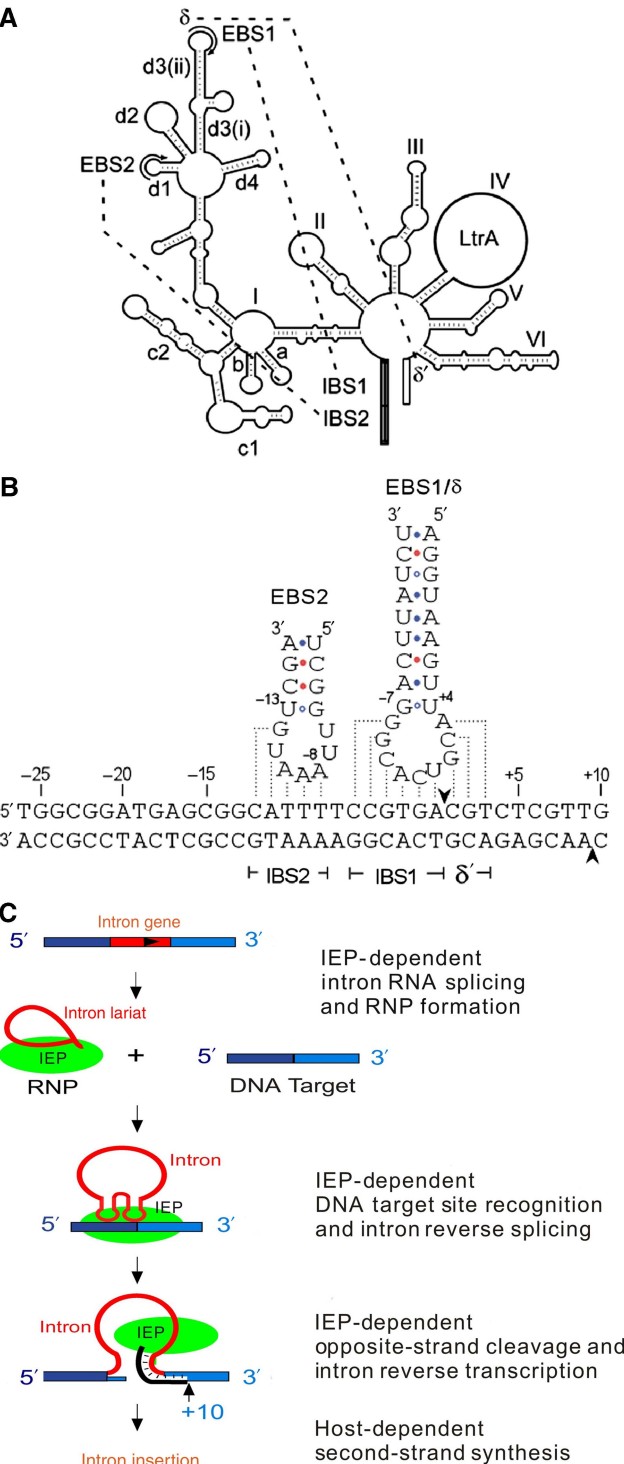

**Figure 1** Targetron structure and mechanism. (**A**) Schematic of the structure of the Ll.LtrB intron, adapted from Perutka *et al* (2004). Domains are labeled, and dotted lines represent contacts made during splicing. For biotechnology applications such as that presented here, the *LtrA* gene (intron-encoded protein) is removed and expressed separately. The *Mlu*I restriction site used for cloning into the intron occupies the location where the intron-encoded protein is found in the wild type. (**B**) Base-pair contacts involved in DNA target-site recognition, using the LtrB.lacZ.635s intron (see Supplementary Table S1) as an example (analogous contacts are made during intron splicing from RNA). Dotted lines show base-pairing interactions, and the arrows show the sites where the DNA is cut. Numbering is relative to the insertion site of the intron, which is between the −1 and +1 bases. (**C**) General mechanism of intron splicing and targeting. RNP, ribonucleoprotein; IEP, intron-encoded protein. While the structure of the EcI5 intron differs from that of Ll.LtrB (Zhuang *et al*, 2009), the target-site base-pairing contacts and overall mechanism are similar.

Figures S1A and B). The linker also determines the specificity of recombination, such that any given *lox* site can only recombine with other *lox* sites having compatible linkers. Many linker mutants with orthogonal specificities are known (Siegel *et al*, 2001; Langer *et al*, 2002). Mutations in the flanking repeats (the 'arms') affect the binding affinity of Cre and can be used to control the direction of recombination. For example, variant *lox*66 and *lox*71 sites are functional but upon recombination with each other form a *lox*72 site that is no longer recognized by Cre (Albert *et al*, 1995) (see Supplementary Figure S1C).

We began by inserting *lox* sites into intron domain IV (the site of the IEP open reading frame (ORF) in wild-type introns) of the *lacZ*-targeting introns LtrB.LacZ.635s and EcI5. LacZ.912s (Supplementary Table S1; the numbers 635 and 912 indicate the position in the *lacZ* gene at which the introns insert, and 's' (as opposed to 'a') indicates that the introns insert into the sense strand of the gene). However, some of the

initial *lox*-site constructs significantly reduced the integration efficiency of the introns. We hypothesized that the tight hairpins that were predicted to be formed by the symmetric lox sites (Zuker, 2003) might disrupt the tertiary structure of the intron, and thereby inhibit splicing or insertion. The *lox*-site inserts were therefore redesigned to include the sequence 'GAGAG' at both ends of the insert to increase the flexibility of the structures, as judged by the presence and size of non-base-pairing regions in the predicted structures, particularly at the base of the stem. This largely restored insertion efficiency (see Figure 2). We hypothesize that this trend occurs because inserts having inflexible structures are more likely to interfere with proper folding of the catalytic structures of the intron than inserts having flexible structures, which can be moved out of the way of other formations.

Statistical analyses (see Supplementary information) confirmed that the inserts fall into two classes: one of wild-type efficiency and one of impaired efficiency. The 2ML5 insert was

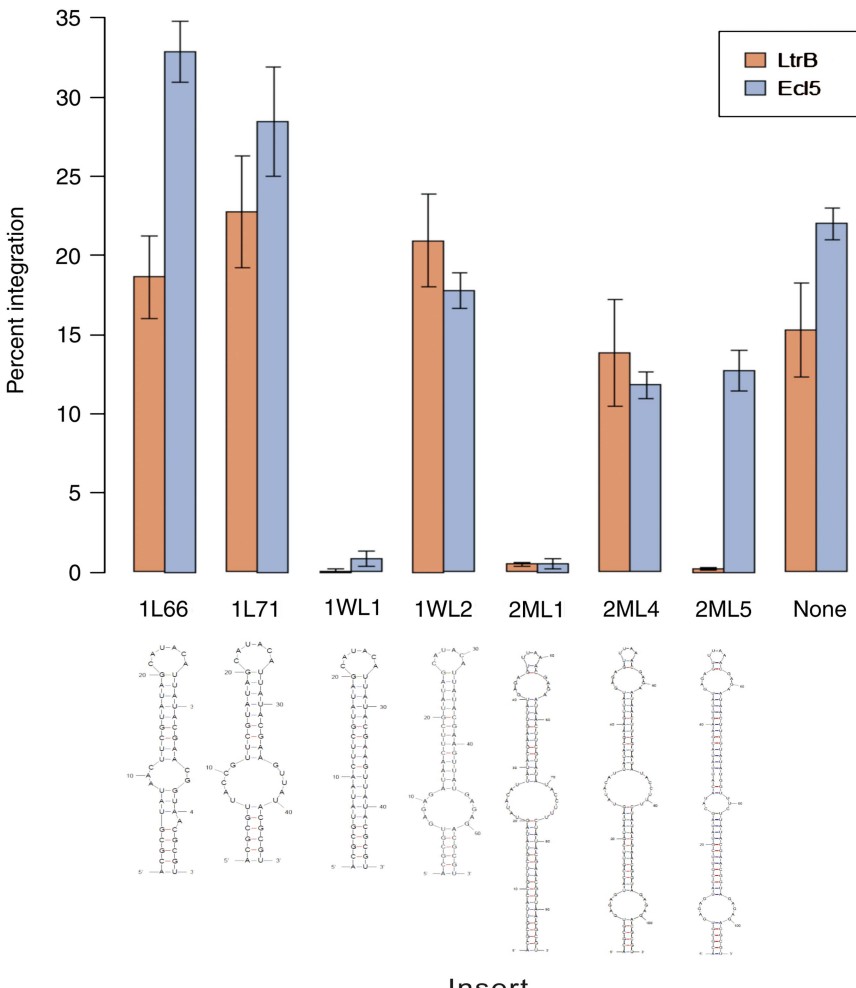

**Figure 2** Effect of *lox* insert on intron efficiency. Different *lox* sequences were inserted into the *Mlu*I site of the LtrB.LacZ.635s (Ll.LtrB) and EcI5.LacZ.912s (EcI5) introns (see Supplementary Table S1), and efficiency was screened by counting the number of white colonies obtained. Error bars are standard error of three replicates, each representing a separate transformation of the intron plasmid into the recipient strain. The identities of the *lox* inserts are as follows, where all sequences are flanked by *Mlu*I sites: 1L66–*lox*66; 1L71–*lox*71; 1WL1–*lox*P (wild-type *lox* site); 1WL2–1WL1 plus a flexible base; 2ML1–*lox*511 with the *lox*71 arm mutation (*lox*511/71) and *lox*FAS with the *lox*66 arm mutation (*lox*FAS/66), separated by a *Pme*I site and a short linker; 2ML4–2ML1 plus a flexible base; and 2ML5–identical to 2ML4 except with *lox*71 and *lox*m2/66 instead of *lox*511/71 and *lox*FAS/66. At the bottom are the RNA structures of the inserts as determined by Mfold (Zuker, 2003).

the only insert that performed markedly differently in the Ll.LtrB intron versus the EcI5 intron. The poor performance of the 2ML5 insert in the Ll.LtrB intron may be due to its relative inflexibility at the central non-base-pairing region as compared with the 2ML4 insert, which is otherwise similar in structure.

## Overview of genomic manipulations of *E. coli* chromosome

Introns were targeted to a variety of insertion sites in *E. coli* (Figure 3); these sites were chosen to flank genomic regions that had previously been shown to be non-essential and amenable to deletion (Kolisnychenko *et al*, 2002; Fukiya *et al*, 2004). A list of introns used in the present work is given in Supplementary Table S1. Figures 4A–D show schematics for using this system to implement insertions, deletions, inversions, and cut-and-paste operations, respectively.

## Insertions (recombination-mediated cassette exchange)

After targetron integration, genomic insertions were performed by recombination-mediated cassette exchange (RMCE), using the EcI5.LacZ.1806s intron to deliver an incompatible pair of *lox*P and *lox*m2 (Langer *et al*, 2002) sites to the genome (Figure 4A). The 1806s intron for inserting into *lacZ* was used for most subsequent modifications instead of the 912s intron due to its higher efficiency, approaching 97% (Zhuang *et al*, 2009). The use of incompatible linker mutations prevents inversion or deletion of the sequence between the *lox*

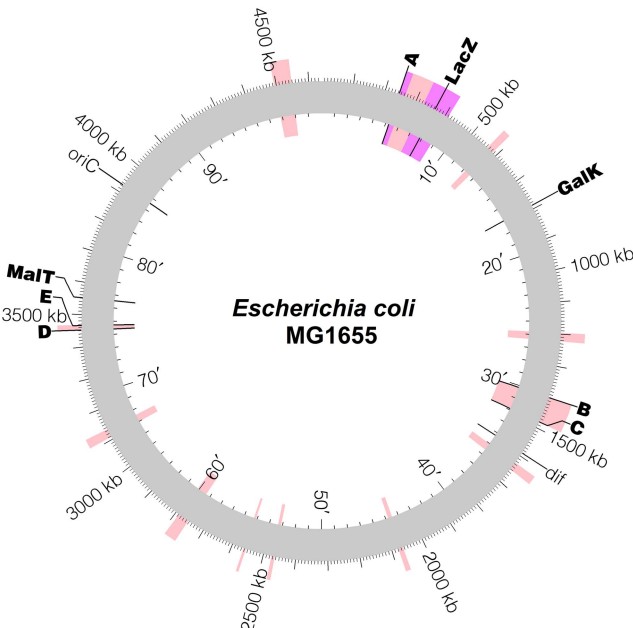

**Figure 3** Genomic integration sites of the introns. Insertion sites of introns used in the present work are labeled in bold type. Pink highlights are regions previously deleted by Kolisnychenko *et al* (2002), and the purple highlight is a region previously deleted by Fukiya *et al* (2004). The intron used for *lacZ* is Eci5.LacZ.1806s (see Supplementary Table S1 online) unless otherwise noted. Image made using Circos (Krzywinski *et al*, 2009).

sites, and the use of arm mutants makes the recombination reaction unidirectional and allows multiple insertions to be made without cross-reactivity. To examine the effect of various experimental parameters (in particular, incubation time, copy number of the delivery plasmid, and strain background) on the efficiency of RMCE, we first delivered a T7 promoter to the genome along with the *lox* sites in the EcI5.LacZ.1806s intron, and separately provided both a promoterless GFPuv gene, flanked by *lox* sites on a pUC19 vector or a pACD vector (derived from pACYC184), and a Cre-expressing plasmid (pQL269; Liu *et al*, 1998). The pUC19 high-copy plasmid is present at about 500 copies per cell (Chambers *et al*, 1988), whereas pACYC is present at only about 20 copies per cell (Chang and Cohen, 1977). In co-transformed cells, GFP expression (*via* the endogenous T7 RNA polymerase) should only occur upon insertion into the genomic target site (see Figure 5A). Two *E. coli* strains, HMS174(DE3) (a K-12 strain related to MG1655) and BL21(DE3) (a B strain), that contained intron-delivered *lox* sites were used and were plated at 1, 2, and 3 days after transformation. Efficiency was gauged by manually counting colonies. The results are shown in Figure 5B, and a statistical analysis of the results is presented in Supplementary information.

In interpreting these results, we first note the significant effect of increasing time on efficiency of insertion. This is likely because interaction between the delivery plasmid and the chromosome occurs at random during any given period, and thus the chance of an interaction occurring increases with time (though in general little is gained by waiting 3 days as opposed to 2). The better performance of the lower-copy vector versus the high-copy vector is surprising at first but may be a result of the lower opportunity for Cre-mediated swapping of cassettes between plasmids in the lower-copy case. The effect of strain, which was not statistically significant except in interaction with other factors, seems to be in modulating the influence of time and copy number. In particular, the effect of time was more pronounced in HMS174(DE3), and the effect of vector copy number was more pronounced in BL21(DE3).

We then further examined the effect of genomic location on RMCE insertion efficiency by repeating the experiment using the lower-copy pACD vector in HMS174(DE3) at two new loci, the *galK* gene and the *malT* gene. The results are shown in Figure 5C. While efficiency of integration into the *malT* locus was worse than at the other loci (see Supplementary information for a full statistical analysis), in all cases the efficiency was high enough by the second day that screening for insertions via colony PCR could be easily performed.

To demonstrate not just the efficiency but the broad utility of this system, we then proceeded to insert the 12-kb DEBS1-TE polyketide synthase operon (Wiesmann *et al*, 1995; Kodumal *et al*, 2004) into the *lacZ* gene of *E. coli* K207-3 (Murli *et al*, 2003). The delivery-plasmid *lox* sites used in earlier experiments were inserted on either side of the operon in the pET26b-DEBS1-TE plasmid using conventional cloning methods. The pET vectors are built on the pBR322 backbone (Rosenberg *et al*, 1987), which is similar in copy number to the pACYC backbone used for the pACD plasmids (Green and Sambrook, 2012). Insertion of the entire operon into the *lacZ* gene was facile (as judged by PCR across an insertion junction) and also showed a trend of increased insertion efficiency as a

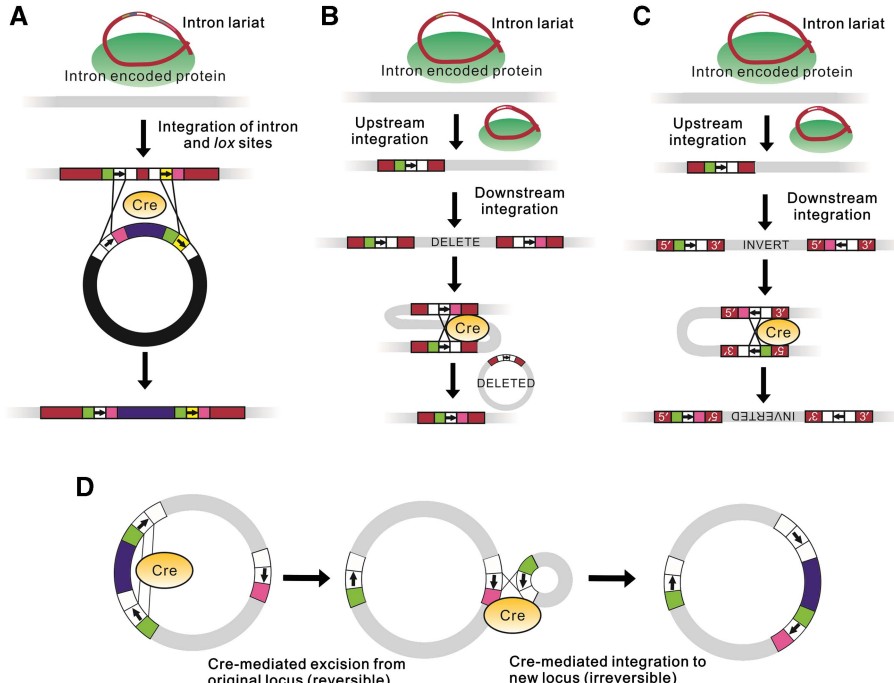

**Figure 4** Genome edits performed. In the figure, *lox* sites are represented by three boxes (arm, linker, and arm), where white represents wild-type *lox*P sequence, green represents the *lox*71 mutant arm, pink represents the *lox*66 mutant arm, yellow represents an incompatible *lox* linker, and the arrows represent the linker orientation. (**A**) Inserting exogenous DNA (recombinase-mediated cassette exchange). Two *lox* sites having incompatible linker regions and differing arm mutations are delivered to the genome using an intron. The sequence to be inserted is then delivered between *lox* sites identical to those in the genome except having opposite arm mutations. The formation of non-functional *lox*72 sites makes the process irreversible. (**B**) Procedure for deleting genomic sequences. A *lox*71 site is carried by an intron upstream of the region to be deleted, and a *lox*66 site is carried downstream. Cre-mediated recombination then deletes the intervening region, leaving a non-functional *lox*72 site behind. (**C**) Procedure for inverting genomic sequences. The procedure is the same as in (**B**), except the *lox* sites have opposing orientations. In the example shown, inverted repeats result from the recombination and would be subsequently deleted by the cell. (**D**) Procedure for one-step cut-and-paste (after placing *lox* sites using introns (not shown)). The first (reversible) step is Cre-mediated deletion, followed by Cre-mediated reinsertion at the target site that is made irreversible by the formation of a *lox*72 site.

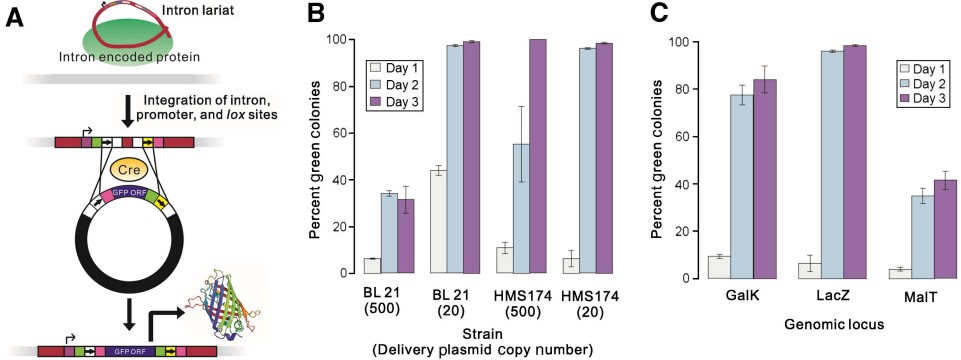

**Figure 5** GFP reporter assay for Cre/*lox*-mediated gene insertion. (**A**) Overview of the method. A T7 promoter is first delivered to the genome with an intron. A promoterless GFP ORF (with ribosome binding site) is then inserted via Cre/*lox*, such that GFP expression is only seen upon insertion. Color-coding as in Figure 4. (**B**) Results as a percentage of green colonies, by strain, delivery-plasmid copy number, and incubation time. Error bars are the standard error of three replicates. On day 3, the HMS174(DE3) (High) colonies were visually homogenous and were thus also assayed by PCR. (**C**) Results as a percentage of green colonies, by genomic location, in HMS174 using the lower-copy vector. The data for *lacZ* are identical to those for HMS174(DE3) (20) in (**B**). Error bars are the standard error of three replicates.

function of incubation time; after 3 days of incubation, 25/25 screened colonies tested positive for the insertion.

One of the features of this manipulation is that insertion also inactivated the *LacY* gene. Thus, IPTG sensitivity is reduced due to a smaller amount being transported into the cell (Mehdi *et al*, 1971), and protein production can be more precisely modulated, resulting in a lower fitness load on the cell. When two of these colonies were screened more fully using

overlapping PCR amplifications that were subsequently sequenced, both were found in fact to contain the entire polyketide synthase operon without error (see Supplementary Figure S2). In principle, insertions of any size could be made at similar efficiency, limited only by the constraints of genome structure (Esnault *et al*, 2007).

We have devised a set of vectors to facilitate the use of RMCE insertion of cassettes into the genome: pX10, pX11, pX20,

and pX21. All the vectors contain the *sacB* gene for counter-selection on sucrose, as well as two T7 terminators between the *sacB* gene and the *lox* sites to prevent read-through from the *sacB* promoter to the delivery target between the *lox* sites. Vectors pX10 and pX11 contain the incompatible *lox* pair of *lox*FAS/71 and *lox*511/66 for use with the 2ML4 pair of *lox* sites (see Figure 2) at the integration site, and vectors pX20 and pX21 contain the incompatible *lox* pair of loxm2/71 and *lox*66 for use with the 2ML5 pair of *lox* sites (see Figure 2) at the integration site. For ease of cloning, vectors pX10 and pX20 contain a *Pme*I restriction site between the *lox* sites, whereas vectors pX11 and pX21 have the multiple cloning site from the pUC vectors between the *lox* sites.

## Deletions

We used the same methods to demonstrate larger scale manipulations of the cellular genome of *E. coli* MG1655(DE3). First, we attempted large-scale deletions (Figure 4B). The *A-lacZ*, *D-E*, and *B-C* regions (see Figure 3) were deleted both sequentially (in the order given) and simultaneously. A set of three mutually incompatible *lox* sites (*lox*P, *lox*2272, and *lox*N; Livet *et al*, 2007) was used for the simultaneous triple deletion. The use of arm mutants once again allowed multiple deletions to be made without cross-reactivity.

Screening for the deletions was performed via colony PCR as depicted in Figure 6A. Three PCR amplifications can be used to characterize each deletion: two amplicons that bridge the genomic sites at which *lox*-carrying targetrons are inserted,

and one amplicon that bridges the expected deletion between those sites. The first two PCRs testing for insertion are expected to give relatively small bands (several hundred base pairs) when performed on the wild-type strain and larger bands (about 1 kb larger) when performed on a strain harboring targetrons at the expected sites. Upon successful deletion of the intervening region, all of these bands should no longer be generated. Instead, the PCR that bridges the two sites should yield a new band of a predicted size. Doing all three PCR amplifications on all three strains (wild-type, wild-type harboring insertions, and recombined (induced with Cre)) provided clear diagnostic signatures of the recombination events. When artifact bands near the sizes of expected bands were observed, these were further analyzed by sequencing to ensure that they did not represent an off-target rearrangement. The predicted sizes of all of the expected amplicons of the present work (as shown in the gels of Figures 6–8 and Supplementary Figures S3–S5) are listed in Supplementary Table S2.

The gels used for verifying the occurrence of these deletions are shown in Figure 6. In particular, Figure 6B shows the three PCR amplifications performed on the three successive, engineered strains that ultimately resulted in a deletion of 121 kb between the *A* and *lacZ* loci. The deletion-bridging amplicon (Au/Ld-I in Figure 6) from the strain that was finally exposed to the Cre protein (*E. coli* MG1655 E1) was sequenced and found to conform to expectations.

As deletions are added, verification becomes more complex but is performed in exactly the same manner. Figure 6C shows the PCR amplifications for verifying the sequential double-

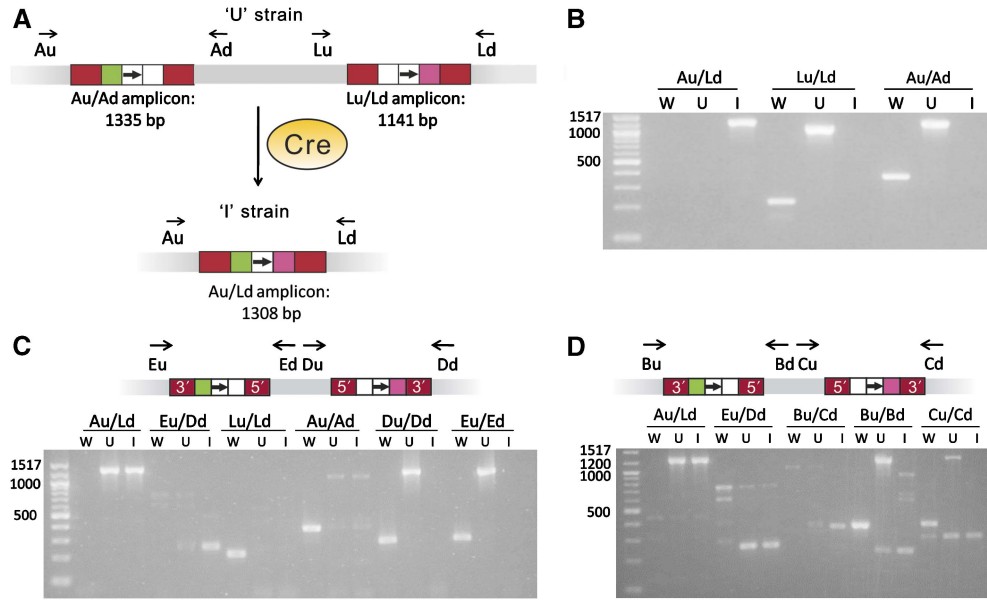

**Figure 6** Verification of genomic deletions. In the figure, 'W' refers to the wild-type *E. coli* strain MG1655(DE3); 'U' refers to the relevant uninduced strain, in which introns and *lox* sites have been placed but Cre has not been added; and 'I' refers to the induced strain, which results from Cre-mediated recombination of the 'U' strain. For primers, the first letter indicates the genomic location the primer amplifies (where 'L' refers to the *lacZ* locus), and the subsequent 'u' or 'd' designates the primer as 'up' or 'down.' PCR products are designated by the two primer names separated by a slash. '5''' or '3''' refers to the sense strand of the intron. (**A**) Methodology, using the deletion of the *A-lacZ* region as an example. (**B**) Verification of the strain (*E. coli* MG1655 E1) containing a deletion of the *A-lacZ* region, as shown in (**A**). (**C**) Verification of the sequential double-deletion strain (*E. coli* MG1655 E6), with schematic corresponding to the 'U' strain. 'U' here is *E. coli* MG1655 E1 with introns inserted to delete the *D-E* region. The Eu/Dd PCR amplifies the *D-E* deletion site (the *D-E* deletion leaves an inverted repeat behind). (**D**) Verification of the sequential triple-deletion strain (*E. coli* MG1655 E10), with schematic corresponding to the 'U' strain. 'U' here is *E. coli* MG1655 E6 with intron insertions for the deletion of the *B-C* region. Bu/Cd amplifies the *B-C* deletion site (the *B-C* deletion leaves behind an inverted repeat).

deletion strain (*E. coli* MG1655 E6) that harbors deletions of both the *A-lacZ* and *D-E* regions. The Eu/Dd-I band for verifying the deletion of the *D-E* region is small because the recombination results in an inverted repeat of intron sequences that is subsequently removed by homologous recombination. Sequencing results confirmed this interpretation. There were some unexpected PCR-amplified bands. These bands, in particular the Eu/Dd-I, Au/Ad-U, and Au/Ad-I bands, were sequenced and found to match genomic sequences from unrelated regions, and not to off-target rearrangements.

Similarly, Figure 6D shows the PCR amplifications for verifying the sequential triple-deletion strain (*E. coli* MG1655 E10) that contains a deletion of the *B-C* region in addition to the *A-lacZ* and *D-E* regions. The unexpected band in the Bu/Cd-U lane was sequenced and found to be from an unrelated genomic region. The Bu/Cd-I amplicon that confirms the deletion also represents the formation and removal of an inverted repeat, which was confirmed by sequencing. The simultaneous triple-deletion strain (E9) and a strain containing a single deletion of the *D-E* region (E11) were verified in the same manner.

As was the case with insertions, the efficiency of deletions approached 100%, with the expected deletion being found in every colony tested. Off-target recombination was rare; in strains designed for deletion, only 1/60 was found to have an inversion when screened after Cre induction; recombination between *lox*72 sites was not detected in any of the modifications reported herein. The removal of inverted repeats upon formation of the *D-E* and *B-C* deletions is interesting in that the *lox* sites are removed entirely and the size of the scar is reduced from hundreds to tens of base pairs.

## Inversions

GETR proved to be robust for other types of recombination that are not easily achieved by other methods. The same methods used to detect deletions can be used to detect inversions, except that four different PCRs are used to verify an inversion: two amplicons bridging the insertions sites and two amplicons bridging the new ends of the inversion (see Figure 7A). Several inversions (see Figure 4C) were executed: namely, between the *A-lacZ*, *B-lacZ*, *E-lacZ*, and *D-E* loci. All colonies screened by PCR soon after addition of Cre tested positive for the expected inversion. Some inversions were only detected immediately after adding Cre and were not detected at later time points. This is in line with previous studies of inversions in the *E. coli* genome, some of which are not well tolerated (Esnault *et al*, 2007). Inversions into the *lacZ* locus were transient when an inverted repeat was formed and subsequently deleted but were stable when non-homologous introns were used, suggesting that the intron sequences may function as a buffer against otherwise deleterious rearrangements at this site.

Recombination back to the original state via homologous recombination of the introns could be detected in some cases but was not seen when non-homologous introns were used. In other words, inversions between *lox* sites in homologous introns may be reversible, but inversions between *lox* sites in non-homologous introns are irreversible. We also tested for the

presence of uninverted chromosomes soon after induction of an irreversible inversion between the *E* and *lacZ* loci. All 10 colonies assayed tested positive for both inverted and uninverted chromosomes, though uninverted chromosomes were not found after restreaking.

Gels for verifying the stable inversions are found in Figure 7. Figure 7B shows the PCRs used to verify an inversion between the *A* and *lacZ* loci, as depicted in Figure 7A. Figure 7C shows an analogous set of PCRs for verifying an inversion between the *D* and *E* loci. As these introns are present in opposite orientations upon integration, inversions can occur via homologous recombination, and this inversion is in fact detected at low levels before the addition of Cre. Similarly, the uninverted (reinverted) state can still be detected after the addition of Cre. These bands were confirmed by sequencing. In those strains where inversion had occurred in the absence of Cre, unrecombined *lox* sites were found, whereas in those strains where inversions back to the wild-type state had apparently occurred after the induction of Cre, recombined *lox* sites were found. These results are consistent with homologous recombination between the introns rather than catalyzed recombination between the *lox* sites. No artifacts were seen in these instances, consistent with the presence of a template that could be amplified by the primers.

Figure 7D demonstrates the difference between using homologous introns (where both introns are of the EcI5 type) versus non-homologous introns (one EcI5 and one Ll.LtrB) for delivering *lox* sites to create substantially identical inversions. When non-homologous introns are used, as in the E5 strain, the inversion is only detected (via the $Lu_0$/Ed-I and $Eu$/$Ld_0$-I bands, which are of the expected size and were confirmed by sequencing) after adding Cre, and reversions back to the original state via homologous recombination were not detected. However, when homologous introns were used, as in the E2 strain, PCR products that should only have been seen upon inversion were also seen in the absence of Cre; and furthermore, the uninverted state could still be detected after the addition of Cre, consistent with homologous recombination between introns. The bands of the sizes expected for recombination events, whether due to Cre-*lox* or homologous recombination, were confirmed by sequencing.

## One-step cut-and-paste

Finally, we used combinations of three *lox* sites to effect unique one-step cut-and-paste reactions (Figure 4D). We use the term 'one-step' because the designated region moves directly from one part of the genome to another upon adding Cre, without the requirement for a stable intermediate, such as a plasmid, to act as a shuttle. In particular, we transferred the *D-E* region to the *B* locus, and the *A-lacZ* region to the *E* locus. Six different PCR amplifications were used to validate a stable cut-and-paste reaction, as shown in Figure 8: three amplicons bridging the three intron integration sites (the left set of three triplets in Figures 8A and B), one amplicon bridging the site of the 'cut' (the fourth set of triplets in Figures 8A and B), and two amplicons bridging the boundaries of the 'paste' region (the fifth and sixth set of triplets in Figures 8A and B). The *D-E* to

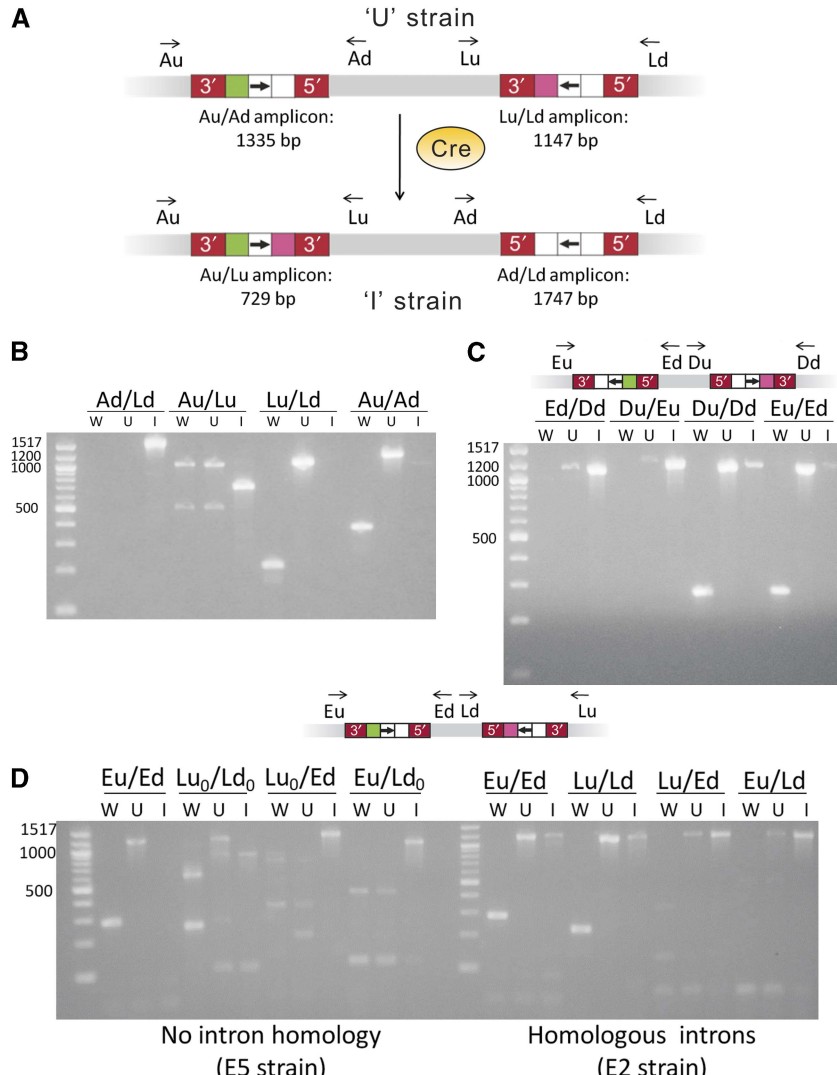

**Figure 7** Verification of genomic inversions. Letter designations are as in Figure 6. (**A**) Methodology, using the inversion of the *A-lacZ* region as an example. '5''' or '3''' refers to the sense strand of the intron. (**B**) Verification of the strain containing an inversion of the *A-lacZ* region (*E. coli* MG1655 E3), as shown in (**A**). (**C**) Verification of the strain containing an inversion of the *D-E* region (*E. coli* MG1655 E4), with schematic corresponding to the 'U' strain. (**D**) Comparison of a strain containing an inversion of the *E-lacZ* region using homologous introns (*E. coli* MG1655 E2) and a strain containing the same inversion using non-homologous introns (*E. coli* MG1655 E5), with schematic corresponding to the 'U' strains. The subscript in $Lu_0$ and $Ld_0$ signifies that these primers amplify the insertion site of the LtrB.LacZ.635s intron rather than the Ecl5.LacZ.1806s intron used elsewhere.

*B* transfer was detected (via the 'cut' and 'paste' bridging amplicons) in every colony soon after exposure to Cre but was not stable upon restreaking.

The *A-lacZ* region was stably translocated to the *E* locus in both possible orientations, with 3/5 and 4/5 colonies positive for the translocations after overnight growth in liquid culture after transformation. Gels for verifying these recombinations can be found in Figure 8. In the case shown in Figure 8A, intron homology allows inversions to occur back and forth between the *lacZ* and *E* loci, but the complete cut-and-paste was only seen upon addition of the Cre protein. Figure 8B shows a similar case where the orientation of the *lox* site at the target (*E*) locus is reversed with respect to the case shown in Figure 8A. The expected rearrangement was obtained, but since the insertion at *E* is in the opposite orientation, inversions and reversions resulting from intron homology

were avoided. The key bands for confirming the cut-and-pastes (Au/Ld-I, Lu/Ed-I, Eu/Ad-I, Eu/Lu-I, and Ed/Ad-I) as well as the bands expected to result from homologous recombination (Lu/Ld-U and Lu/Ed-U in Figure 8A) were confirmed by sequencing.

## Growth of *E. coli* strains with chromosomal rearrangements

A summary of the genomic rearrangements generated in *E. coli* is given in Table I, and a list of *E. coli* strains containing these rearrangements is given in Table II, along with doubling times measured for these strains as a proxy for fitness. A statistical analysis of the doubling times indicated that the strains fall broadly into two groups, one group having approximately

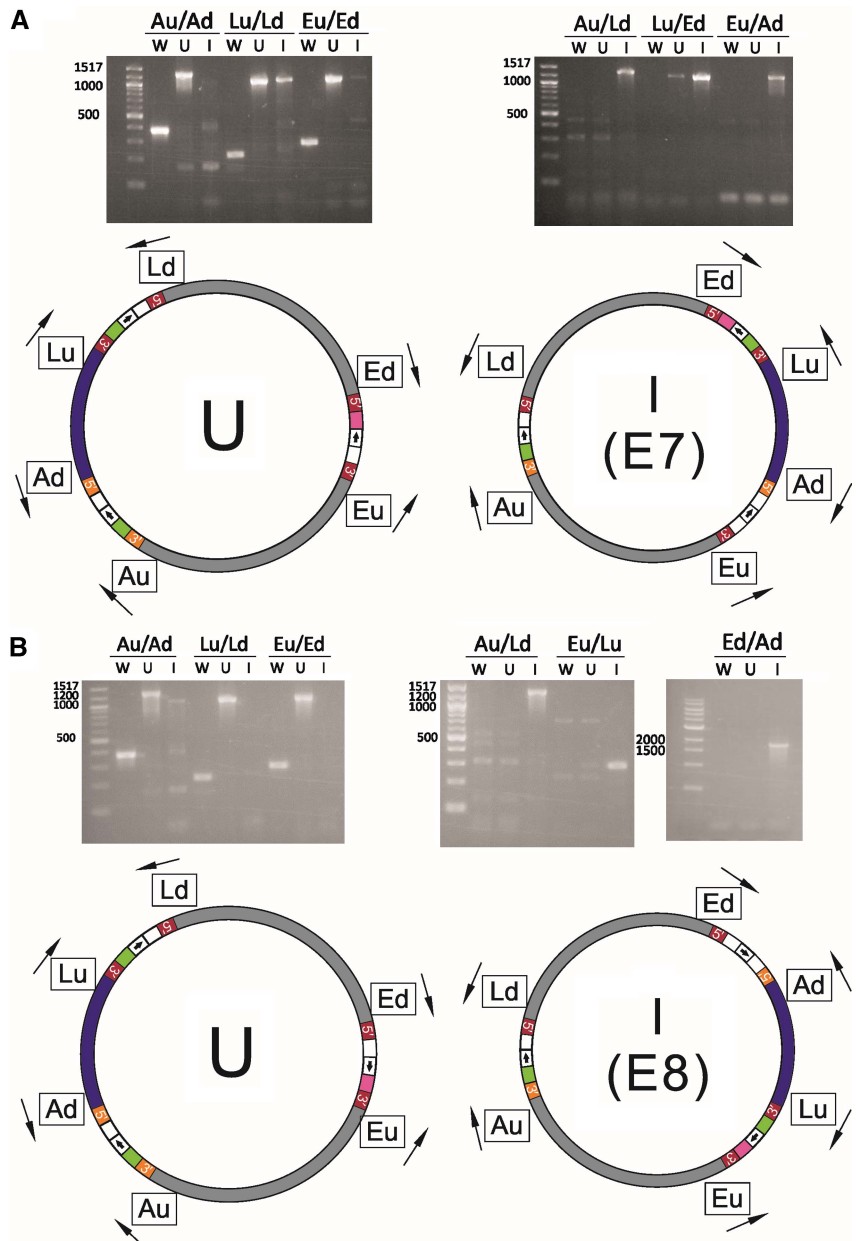

**Figure 8** Verification of one-step cut-and-pastes. Orange is Ll.LtrB intron sequence, which is non-homologous with respect to EcI5 intron sequence shown in red. Letter and number designations are as in Figure 6. (**A**) Verification of the strain (*E. coli* MG1655 E7) containing a cut-and-paste (translocation) of the *A-lacZ* region to the *E* locus in the reverse orientation. (**B**) Verification of the strain (*E. coli* MG1655 E8) containing a cut-and-paste (translocation) of the *A-lacZ* region to the *E* locus in the forward orientation.

wild-type growth, and the other group having impaired growth (see Supplementary information). All strains that either lacked the *A-lacZ* region or had an inversion between the *E* and *lacZ* regions showed significantly impaired growth. Interestingly, the strains containing a cut-and-paste of the *A-lacZ* region to the *E* site displayed wild-type growth rates.

## Genome engineering in diverse bacteria

While GETR is obviously broadly useful for creating virtually any type of rearrangement, the real utility of the method

appears when moving beyond *E. coli* as a model system. We therefore applied the method to make genomic modifications in three additional phylogenetically diverse species: *Staphylococcus aureus*, *B. subtilis*, and *S. oneidensis*.

The Gram-positive bacterium *Staphylococcus aureus* is an intensely studied human pathogen, and the rise of drug-resistant strains in recent years has given new urgency to the development of prophylactic and therapeutic approaches to treatment (Otto, 2012). We therefore attempted to delete the 15-kb *Staphylococcus aureus* pathogenicity island 1 (SaPI-1) from *Staphylococcus aureus* RN10628 (Ubeda *et al*, 2009), to create a strain that might serve as a live vaccine. Highly

**Table I** Summary of intragenomic rearrangements in *E. coli*

| Loci recombined[a] | Largest distance between recombined loci | Type of modification | Inverted repeat generated | Stability[b] | Recombination in the absence of Cre[c] |
|---|---|---|---|---|---|
| *A-lacZ* | 121 kb | Deletion | No | Stable | Yes |
| *A-lacZ* | 121 kb | Deletion | No | Stable | No |
| *B-C* | 81 kb | Deletion | Yes | Stable | No |
| *D-E* | 14 kb | Deletion | Yes | Stable | No |
| *A-lacZ* | 121 kb | Inversion | Yes | Unstable | No |
| *A-lacZ* | 121 kb | Inversion | No | Stable | No |
| *B-lacZ* | 1 Mb | Inversion | Yes | Unstable | No |
| *E-lacZ* | 1.5 Mb | Inversion | No | Quasi-stable | Yes |
| *E-lacZ* | 1.5 Mb | Inversion | No | Stable | No |
| *D-E* | 14 kb | Inversion | No | Quasi-stable | Yes |
| *A-lacZ* to *E* | 1.5 Mb | Cut-and-paste | Yes | Stable | Some |
| *D-E* to *B* | 2.1 Mb | Cut-and-paste | Yes | Unstable | Some |

[a]The LtrB.LacZ.635s intron was used in the first *A-lacZ* deletion (with recombination in the absence of Cre) and in the stable *A-lacZ* and *E-lacZ* inversions; the EcI5.LacZ.1806s intron was used in all other cases (see Supplementary Table S1 online).
[b]'Stable' means the recombination remained present unchanged through multiple rounds of regrowth. 'Unstable' means the recombination was detected initially but was not detected after multiple rounds of regrowth. 'Quasi-stable' means the recombination was still detected after multiple rounds of regrowth, but back-recombination due to homologous recombination was also detected.
[c]Recombination was seen only in the absence of Cre when (1) the introns were homologous and (2) the introns were oriented so as to allow the homologous recombination to occur. 'Some' recombination in the absence of Cre for the cut-and-pastes refers to the fact that inversions caused by homologous recombination were detected, but the complete cut-and-paste did not occur without Cre.

**Table II** Doubling times of *E. coli* strains with intragenomic rearrangements

| Strain | Doubling time | Standard error[a] | Description |
|---|---|---|---|
| MG1655 | 24.94 | 0.1 | |
| MG1655(DE3) | 24.44 | 0.2 | Base strain for E1-E11 |
| E1 | 28.33 | 0.5 | A-lacZ deletion |
| E2 | 33.25 | 0.6 | E-lacZ inversion (reversible) |
| E3 | 22.03 | 0.1 | A-lacZ inversion (irreversible) |
| E4 | 24.65 | 0.4 | D-E inversion (reversible) |
| E5 | 30.5 | 1.2 | E-lacZ inversion (irreversible) |
| E6 | 33.94 | 0.8 | lacZ-A, D-E deletion |
| E7 | 25.24 | 0.5 | lacZ-A region to E, reverse orientation |
| E8 | 23.65 | 0.4 | lacZ-A region to E, forward orientation |
| E9 | 29.27 | 1.0 | lacZ-A, D-E, B-C simultaneous deletion |
| E10 | 31.39 | 0.2 | lacZ-A, D-E, B-C sequential deletion |
| E11 | 24.45 | 0.5 | D-E del |

[a]Error is from three replicates.

efficient Ll.LtrB-type introns were generated that could integrate in the first gene (*int*) of SaPI-1 and also downstream of the pathogenicity island. After transformation of the Cre-expressing plasmid pRAB1 (Leibig *et al*, 2008), 40/40 colonies tested contained cells harboring the expected deletion, and 19/40 colonies tested still harbored SaPI-1. The deletion was detected and verified via PCR and sequencing (see Supplementary Figure S3). No chromosomes containing SaPI-1 were detected in restreaked colonies. The deletion was stably maintained.

*Bacillus subtilis* is a model system for the study of Gram-positive bacteria, including studies on sporulation (Earl *et al*, 2008; Higgins and Dworkin, 2012). We designed and built two Ll.LtrB-type introns that inserted into the sense strands of the *sacB* and *yhcS* (*srtA*) genes of *B. subtilis* at efficiencies 98 and

91%, respectively (Yao, 2008; Whitt, 2011). We used these introns to deliver a *lox*71 site to the *sacB* locus and a *lox*66 site to the *yhcS* locus in *B. subtilis* 168, positioning the intervening region for inversion. Upon addition of the Cre-expressing plasmid pCrePA (Pomerantsev *et al*, 2006), 4/11 screened colonies tested positive for the inversion via colony PCR (Supplementary Figure S4). Sequencing of the PCR products gave a sequence consistent with the expected inversion. The inversion covers about 1.5 Mb of the 4.2 Mb genome and was not seen upon restreaking, indicating that it was not well tolerated.

*Shewanella oneidensis* is a Gram-negative bacterium that is a model system for extracellular electron transfer, with potential applications in bioremediation and energy (Fredrickson *et al*, 2008). We designed an Ll.LtrB targetron that inserted into the ribosomal *rrs* genes (*rrsA* through *rrsI*) in *S. oneidensis* (see Supplementary Figure S5A). This targetron included a *loxP* site in the intron to facilitate subsequent genomic rearrangements, was cloned onto the broad-host range plasmid RP4, and was introduced into *S. oneidensis* via conjugation. We screened single colonies for insertions at each site by PCR using one primer complementary to the intron and another primer complementary to a unique chromosomal sequence near the insertion site. We initially found insertions in all copies of the *rrs* gene except *rrsC* (Supplementary Figure S5B); subsequent PCRs using other primers to more specifically detect insertions in *rrsC* did yield bands. We found the same pattern of PCR bands after growing one of the colonies overnight in liquid culture, freezing at −80°C, restreaking, and repeating the PCR amplifications on one of the resultant colonies (Supplementary Figure S5C).

# Discussion

As synthetic biology continues to advance, there will be an increasing emphasis on the genome as the unit of engineering,

which allows much larger swaths of DNA to be manipulated than is possible with plasmid-based methods and enhances our ability to study the structure and function of genomes. This is already evidenced by the synthesis and transplantation of whole genomes by the Venter Institute (Lartigue *et al*, 2007; Gibson *et al*, 2008; Lartigue *et al*, 2009; Gibson *et al*, 2010) and the development of technologies such as MAGE that can site-specifically perturb multiple sites in a genome (Wang *et al*, 2009; Isaacs *et al*, 2011). However, both technologies are still time and resource intensive and are currently limited to a relatively small number of organisms (Enyeart and Ellington, 2011).

We have therefore combined the well-known Cre-*lox* recombinase system and the adaptable targetron technologies to create a method we dub GETR. GETR presents several advantages in comparison with recombineering and related methods that make use of the lambda Red functions. One of these advantages is the fact that GETR is very efficient, and, while the use of selectable markers is required for temporary plasmid maintenance, markers need not be used for selecting or maintaining genomic modifications. Recombineering using single-stranded DNA is simple to execute and useful for making small changes such as point mutations, but percent efficiencies are typically in the single digit range in mutator strains (specifically, *mutS* mutants) and are much lower in wild-type strains. Even then, the efficiency of inserting a sequence as large as a *lox* site is in the neighborhood of 1%, and the efficiency of deleting 10 000 bases of genomic sequence is ~0.1% (Wang *et al*, 2009). The use of the full complement of lambda Red proteins allows larger pieces of double-stranded DNA to be inserted (Datsenko and Wanner, 2000), but selectable markers are typically required and the size of possible insertions is limited to several thousands of bases. Manipulations such as inversions and cut-and-paste operations are also impossible using these methods alone.

Another advantage is that targetrons function at high efficiency in many bacterial strains and thus provide an appealing alternative to recombineering functions in many contexts. While the lambda Red system has been used outside of *E. coli*, it typically does not function as well in other organisms and in such cases generally requires 500 nucleotides of target-site homology on either side of the integration cassette to obtain reliable results (Beloin *et al*, 2003; Derbise *et al*, 2003; Rossi *et al*, 2003; Lesic and Rahme, 2008; Jia *et al*, 2010). This requires significantly more labor than the 30–40 nucleotides of homologous sequence required in *E. coli*. In *Pantoea ananatis*, the system only worked well after selection of mutants resistant to the toxic effects of the lambda Red proteins (Katashkina *et al*, 2009). In some organisms, such as *Pseudomonas syringae* (Swingle *et al*, 2010) and *Mycobacterium tuberculosis* (van Kessel and Hatfull, 2007, 2008), alternative recombineering functions have been discovered, but these do not exceed 0.1% efficiency without selection and also typically require at least 500 nucleotides of homology on either side for reliable results with selectable markers. Recombineering using single-stranded oligonucleotides for making point mutations has been reported in *Lactobacillus* species, but the electroporation of 100 μg of DNA (1,000 times the optimal amount in *E. coli*) was required for efficient mutagenesis

(van Pijkeren and Britton, 2012). Wang *et al* (2012) were also able to demonstrate gene disruption in *B. subtilis* using single-stranded DNA, but the method required the use of selectable markers and the generation of single-stranded DNA long enough to encode those markers. Datta *et al* (2008) have identified a number of other possible recombineering proteins from a variety of species, but to our knowledge none of these have yet been demonstrated as recombineering tools in their natural hosts.

Another common method of genome engineering is the use of suicide plasmids. For instance, temperature-sensitive integrable plasmids have been developed for all the systems described here other than *S. oneidensis* (Hamilton *et al*, 1989; Luchansky *et al*, 1989; Biswas *et al*, 1993; Link *et al*, 1997), and systems based on plasmids requiring expression of the *pir* (Kolter *et al*, 1978; Miller and Mekalanos, 1988) or *repA* (Leenhouts *et al*, 1996) genes to replicate have also been frequently employed. These systems are most useful for gene replacements. For the types of modifications discussed in the present work, suicide plasmids present many of the same limitations as recombineering, such as requirements for selectable markers and large regions of homology, and are limited by poor efficiency and relatively high background. The profusion of research into alternative recombineering systems in recent years, described above, is symptomatic of broader dissatisfaction with suicide plasmids as genetic tools, and the present system represents a favorable alternative to suicide plasmids for large-scale genomic modifications.

A more recent addition to the set of tools available for genome engineering is the CRISPR/Cas9 system, which adapts the site-specific RNA-mediated restriction system of bacteria toward making targeted double-strand breaks in genomic DNA (Cho *et al*, 2013; Cong *et al*, 2013; Hwang *et al*, 2013; Mali *et al*, 2013). Methods of genome engineering relying solely on the creation of double-strand breaks have not traditionally gained much traction in bacterial systems. Besides the requirement for selectable markers, the efficiency of double-strand break repair tends to be poor in bacteria, since most prokaryotes are only capable of repairing breaks via homologous recombination, and those that can carry out non-homologous end joining have only a rudimentary system for doing so (Aravind and Koonin, 2001; Hefferin and Tomkinson, 2005). CRISPR-Cas9-mediated cutting of genomic DNA has been shown to be lethal to bacteria (Bikard *et al*, 2012), but Jiang *et al* (2013) have recently reported that this method can be used to select for the integration of mutated DNA homologous into the cut site.

However, the CRISPR/Cas9 system alone is only of functional efficiency in bacteria that have very active recombination systems, such as *Streptococcus pneumoniae*, and in those systems the CRISPR-Cas9 expression construct must also be integrated into the genome along with a selectable marker and then subsequently removed. In *E. coli*, the CRISPR-Cas9 system has been shown to increase the efficiency of recombineering by cleaving the genome at unmodified sites (and thereby selecting for modified strains), but this method also has the inherent limitations of recombineering; that is, the requirement of a mutator strain for high efficiency, limitation to relatively small changes, and generally poor efficiency in systems other than *E. coli*. That said, it is

possible that a more general application of CRISPR-Cas9 could be to increase the efficiency of targetron-mediated mutagenesis. Finally, recent work by Fu *et al* (2013) demonstrates extensive off-target mutagenesis by CRISPR-Cas9, often at efficiencies comparable to the degree of on-target mutagenesis.

We have demonstrated the utility of targetron-delivered *lox* sites by deleting up to 120 kb of the *E. coli* genome and 15 kb of the *S. aureus* genome, inverting up to 1.5 megabases (one-third) of both the *E. coli* and *B. subtilis* genomes, and stably translocating 121 kb of the *E. coli* genome to another locus 1.5 megabases away. Efficiencies of the Cre-mediated recombinations are typically near 100%. This method compares favorably with another recently reported method for using targetrons to make genomic deletions (Jia *et al*, 2011) that relied on homologous recombination between introns and reported an efficiency of 2/648 for the deletion of a two-gene operon, requiring seven rounds of growth and transfer to new media.

The use of the Cre/*lox* system allows large pieces of foreign DNA to be integrated into genomes at high efficiency. An initial recombination occurs between a *lox* site on the plasmid and a *lox* site in the genome, serving to integrate the entire plasmid into the genome, and a second recombination event then occurs between the other two *lox* sites and removes the plasmid sequence. We found no evidence for a difference in efficiency between inserting 1 and 13 kb into the *E. coli* genome via RMCE. Given the high efficiency observed during the construction of large deletions and inversions, the limiting factor in RMCE would thus seem to be the initial encounter between the plasmid and the genome, and not the size of the insertion. The speed and efficiency of the second recombination event is presumed to be rapid and essentially 100% efficient, similar to the other intragenomic recombinations we report.

While large-scale inversions were presented here primarily as a demonstration of the lack of size limits for generating rearrangements using our method, artificial inversions have traditionally been used for studying genome structure and its constraints (Hill and Gray, 1988; Rebollo *et al*, 1988; Segall *et al*, 1988; Guijo *et al*, 2001; Campo *et al*, 2004; Garcia-Russell *et al*, 2004; Valens *et al*, 2004; Esnault *et al*, 2007), and the approaches presented herein allow such studies to be more easily performed in many more systems.

The one-step cut-and-paste method we present is of particular interest given that it allows one piece of a genome to be inserted within another site, without the accumulation of intervening intermediates, an operation that is not possible with any other technique. The cut-and-paste method could also be applied to more nuanced studies of genome structure constraints. For instance, the effect of moving different structural domains or of swapping two domains, such as the Ori and Ter domains, could be examined. Additionally, expression levels tend to be dependent on genomic location, with, for instance, genes nearer the origin tending to be more highly expressed (Cooper and Helmstetter, 1968; Rocha, 2008); and thus, cut-and-pastes could be used as a simple means for modulating the overall expression levels of super-operons (Lathe *et al*, 2000; Rogozin *et al*, 2002) or other large genetic units. The ability to move DNA between species without regard for inherent similarities or phylogenetic relationships opens up the possibility of using genomic editing for rapidly adapting bacterial genomes.

Targetron genomic engineering technology can be readily practiced by almost any laboratory. The algorithm for retargeting the targetrons is available online. The targeting sites in the intron can be changed via restriction cloning of a short fragment of DNA that can be created via two PCRs or synthesized in its entirety (see Materials and methods), followed by the typical time required for ligation, transformation, and sequence validation. Retargeting and the addition of *lox* sites can be performed for multiple introns in parallel. Following electroporation into the target strain, intron induction requires only 1 day, and plated induction colonies grow after 1 day. The method is similar in complexity to *lox*-site placements with lambda Red, but is an improvement on recombineering in that no selectable markers are required and it can be used in strains where lambda Red performs poorly. Similarly, Cre-mediated recombination requires 1 day for electroporation of the Cre-expressing plasmid (and, for RMCE insertions, the delivery plasmid), and 1–2 days for the cells to grow and for recombination to occur. Though we used plasmids (one plasmid carrying the targetron, one plasmid carrying the Cre gene, and, as necessary, a plasmid or other vector carrying DNA to be integrated, delivered by electroporation or conjugation) to deliver targetrons in the present study, phage, direct electroporation or other methods could potentially be used, as well.

The scars left by the GETR method and the possibility of unplanned homologous recombination between introns are potential drawbacks, but we have shown that these can be avoided by careful planning. If intron and *lox*-site orientations are designed so that inverted repeats form upon Cre-mediated recombination, then the repeats will be deleted by the cell, removing most of the intron scar. However, the fact that certain inversions into the *lacZ* locus were viable when a scar was present but not when the scar was removed indicates that such scars may serve as a buffer against deleterious genomic rearrangements. Unwanted homologous recombination between introns can be prevented by the use of non-homologous introns (EcI5 and Ll.LtrB), or by targetron-mediated disruption of the *recA* gene.

Removing the genome-modifying plasmids was also simple. Except in the case of *S. oneidensis*, which required the continued presence of the IEP to allow the intron to splice out from the rRNA genes, a significant fraction (at least 1/3) of colonies were found to have lost the intron-expressing plasmid after the induction process. The Cre-expressing plasmids employed all contained temperature-sensitive origins of replication, and the delivery plasmids for RMCE had the *sacB* gene for counter-selection on sucrose, allowing these plasmids to be easily removed, as well.

In summary, GETR is a new method for genome engineering that can be adapted for use in a variety of bacteria with minimal modifications and without a significant loss of functionality. Large, specific, and varied changes can be made with high efficiency. This approach presents certain advantages over recombineering, particularly when working in strains not closely related to *E. coli*, or when the use of selectable markers is impractical or undesirable. In the case of *Staphylococcus aureus* in particular, recent work has made it

possible to transform clinical strains (Corvaglia *et al*, 2010), opening the way to genome editing of otherwise drug-resistant bacteria to create vaccine strains. As concerns about increasing drug resistance of pathogenic bacteria continue to mount, such strains may prove to be a viable alternative to antibiotics. We also expect the system to be of general utility to synthetic biologists looking to engineer entire genomes, particularly those looking to work in systems other than *E. coli*.

# Materials and methods

Details of plasmid construction can be found in Supplementary Methods, and lists of the introns, plasmids, strains, and oligomers used in the present work are in Supplementary Tables S2–S5.

## Intron retargeting

Introns were designed as described elsewhere (Perutka *et al*, 2004; Zhuang *et al*, 2009). The algorithm is available at http://www.targe-trons.com. Ll.LtrB-type introns were retargeted according to the Sigma-Aldrich User Guide for the TargeTron Gene Knockout System (http://www.sigmaaldrich.com/etc/medialib/docs/Sigma/General_Information/targetron-user-guide.Par.0001.File.tmp/targetron-user-guide.pdf), except that the primers were prepared differently to improve the yield of the PCR amplification. Specifically, 1 μl each of 20-μM solutions of the EBS2 and EBS2AS primers were diluted into 26 μl of water. In all, 2 μl of this mixture and 1.4 μl each of 20-μM solutions of the IBS and EBS1 primers were used in the PCR amplification. The rest of the protocol was not substantially different from the Sigma-Aldrich protocol. Alternatively, the entire retargeted *Hin*dIII/*Bsr*GI fragment was ordered as a gBlock from IDT and cloned directly into the introns to be retargeted.

For the EcI5 introns, two different PCR amplifications were first executed using the IBS1/2S and EBS2AS primers in one reaction, and the EBS1S and EBSR primers in the other. In these reactions, 2 μl each of 10-μM solutions of the two primers and at least 5 ng template (an EcI5 intron having the proper base at the +1 position) were used in 50 μl. The products were subjected to PCR clean-up, and then at least 5 ng of each was combined for use as the template of a second PCR amplification similar to the first except doubled to a total volume of 100 μl, with 8 μl of 10-μM EBSR and 2 μl of 10-μM IBS1/2S as the primers. The product was subjected to PCR clean-up, digested with *Ava*II and *Xba*I, and ligated into the EcI5 vector (having the proper base at the +1 position) cut with *Ava*II and *Xba*I.

## Intron induction

In *E. coli* strains, cells transformed with the intron-expressing plasmid were grown overnight at 37°C in Luria-Bertani (LB) broth plus 34 μg/ml chloramphenicol, diluted to an $OD_{600}$ of 0.05 in 5 ml of LB plus 34 μg/ml chloramphenicol, and then grown for 1 h at 37°C. In all, 250 μl of that culture was then inoculated into 5 ml of LB containing 200 μM IPTG (no antibiotic) and grown for 20 min (for Ll.LtrB-type introns) or 3 h (for EcI5-type introns) at 37°C. (EcI5 introns can also be induced for the shorter time period, but efficiency is somewhat better using the longer period.) The cultures were then put on ice, and 50 μl of a 100× dilution (for Ll.LtrB-type introns) or 1000× dilution (for EcI5-type introns) was then streaked on LB plates (non-selective) pre-warmed to 37°C. The plates were then incubated overnight, and intron integration was screened using colony PCR. A subset of positive colonies was then screened for loss of antibiotic resistance to indicate the absence of the intron-expressing plasmid.

Intron induction in *S. aureus* RN10628 and *B. subtilis* 168 was performed as described elsewhere (Yao *et al*, 2006). Tryptic soy broth (TSB) was used as the growth medium for *S. aureus*, and LB broth was used for *B. subtilis* (with 5 μg/ml erythromycin) and *S. oneidensis* (with 50 μg/ml kanamycin). The T5.rDNA.798s.1WL2R intron was not formally induced.

## Induction of Cre-mediated recombination

For intramolecular recombinations in *E. coli*, the plasmid pQL269 (Liu *et al*, 1998) was electroporated into cells that were then plated on LB plus 100 μg/ml spectinomycin and grown at 30°C until colonies appeared. Occurrence of recombination was screened using colony PCR, and a subset of positive colonies were restreaked on LB (non-selective) and grown overnight at 42°C to cure the plasmid. Freezer stocks were made from these cells, and the analyses shown in Figure 6 through 8 and Supplementary Figure S3 were performed on cells streaked from these stocks. The procedure was essentially the same in *S. aureus* RN10628, except that the cells were electroporated with pRAB1 (Leibig *et al*, 2008) and grown initially on tryptic soy agar (TSA) plus 10 μg/ml chloramphenicol. *B. subtilis* 168 was electroporated with pCrePA and grown on LB plus 5 μg/ml erythromycin.

## Cre-mediated genomic insertion (RMCE)

To assay insertion efficiency, delivery plasmids pACDX3S-GFP and pUC19X3S-GFP were used. These plasmids contain the GFP ORF flanked by T7 terminators and the *lox*71 and *lox*m2/66 sites. Each of the GFP delivery plasmids was transformed into *E. coli* BL21(DE3) Gold and *E. coli* HMS174(DE3) having a T7 promoter, as well as *lox*66 and *lox*m2/71 sites complementary to those in the delivery plasmids, integrated at the *lacZ*, *galK*, or *malT* locus. The Cre-expressing plasmid pQL269 was transformed into the strains, which were then grown at 30°C in liquid culture. At days 1, 2, and 3 after transformation of pQL269, aliquots from each culture were spread on LB plates. The plates were grown overnight at 37°C, and then imaged using a UV backlight and a B&W 061 dark-green filter. Identical strains lacking pQL269 were used as negative controls at each time point. Green colonies were counted manually to determine the insertion efficiency. The entire 3-day procedure was performed three times separately.

The insertion of DEBS1-TE was performed similarly, using pET26b-DEBS1TE-i as the delivery plasmid. Insertion was assayed by colony PCR using primers flanking the 5′ end of the insertion 3 days after transformation of pQL269. Selected positive clones were then further assayed by overlapping PCRs covering the entire operon after removal of the delivery plasmid.

## Doubling time measurements

Overnight cultures of the strains to be measured were diluted in LB to an $OD_{600}$ of 0.001, and triplicates of 500 μl of that culture were placed in a 96-well plate (Nunc). All other wells (including all wells on edges) were filled with 500 μl of sterile media (LB). Growth was measured using a plate reader (Bio-Tek PowerWave 340), pre-heated to and maintained at 37°C with a shaking intensity of 4 for 540 s at a time, with measurements taken every 560 s.

The results were plotted as $\log_2(OD_{600})$ versus time (min). To select the linear region of the curve, each point was assigned a correlation coefficient $R^2$ corresponding to the value of $R^2$ for the line consisting of that point and the three points before and after. Since variance was lower when the same time window was used for all three replicates, the resulting $R^2$ values were averaged for all three replicates at each time point. The longest stretch in which all these averaged $R^2$ values were ⩾0.99 was taken as the linear range. The slope of the least-squares linear fit of each replicate in that time range was then taken as the doubling time.

## Data availability

GenBank accession numbers for the plasmids pX10, pX11, pX20, and pX21, are KF155402, KF155403, KF155404, and KF155405, respectively.

## Supplementary information

# Acknowledgements

We thank S Elledge (Harvard University) for the gift of plasmid pQL269, R Bertram (University of Tübingen) for the gift of plasmid pRAB1, G Christie (Virginia Commonwealth University) for the gift of strain RN10628, S Leppla (National Institutes of Health) for the gift of the plasmid pCrePA, and Andrew Harper (University of Texas at Austin) for constructing plasmid pET26b-DEBS1TE. This material is based upon work supported in part by the National Science Foundation Graduate Research Fellowship under Grant No DGE-1110007, the National Security Science and Engineering Faculty Fellowship (FA9550-10-1-0169), and the Welch Foundation (F-1654). Work in AML's laboratory was supported by NIH grant GM37949 and Welch Foundation grant F-1607.

*Author contributions:* SMC and PJE performed the GFP and DEBS-TE insertions, which were planned by PJE, SMC, ATK-C, and ADE and analyzed by SMC and PJE. JP designed the introns and planned the *S. aureus* deletion. MND constructed most of the introns, and designed, performed, and analyzed the simultaneous triple deletion. JY, JTW, and AML designed and tested the *B. subtilis* introns, and EMQ designed and tested the *S. oneidensis* rDNA intron. PJE and ADE designed the rest of the experiments, and PJE performed and analyzed all other experiments. All authors participated in writing, revising, and editing the paper.

# Conflict of interest

Targetron technology is subject to issued US and foreign patents and patent applications that are licensed by the Ohio State University and the University of Texas to InGex, LLC, which sublicenses the technology to others for commercial applications. JP, AML, the Ohio State University, and the University of Texas are minority equity holders in InGex, LLC, and AML serves as an advisor to InGex, LLC. JP and AML may receive royalty payments for commercial use of the technology. JP is the founder of a company, Targetronics, which sublicenses targetron technology from InGex, LLC.

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
