## [Review Process File · Molecular Systems Biology]

Generalized bacterial genome editing using mobile group II introns and Cre-lox

Peter J. Enyeart, Steven M. Chirieleison, Mai N. Dao, Jiri Perutka, Erik M. Quandt, Jun Yao, Jacob T. Whitt, Adrian T. Keatinge-Clay, Alan M. Lambowitz, Andrew D. Ellington

Corresponding author: Andrew D. Ellington, University of Texas at Austin

Review timeline:	First Submission:	13 December 2011
	Editorial Decision:	01 February 2012
	Second Submission:	31 May 2013
	Editorial Decision:	07 July 2013
	Revision received:	20 July 2013
	Accepted:	23 July 2013

Editors: Andrew Hufton / Maria Polychronidou

Transaction Report:

1st Editorial Decision

01 February 2012

Thank you again for submitting your work to Molecular Systems Biology. We have now heard back from the three referees who agreed to evaluate your manuscript. As you will see from the reports below, the referees find the topic of your study of potential interest. They raise, however, substantial concerns on your work, which, I am afraid to say, preclude its publication in its present form.

The reviewers, in general, agreed that this work describes a genome editing method that may have advantages over existing technologies. Reviewers #1 and #3, however, were not yet fully convinced of the generality or portability of this method, and wrote that tests on additional bacterial species, or at another genomic loci, would substantially improve this work. These two reviewers both rated the "validity of conclusions" and the "suitability of publication" as "medium" in our online system, indicating that additional evidence appears to be required to conclusively demonstrate the broad usefulness of this methodology.

In addition, the reviewers had a number of concerns related to the presentation of this work, and felt that a more detailed description of the experimental methods and plasmids would be essential. Moreover, the last reviewer felt that greater clarity was required in describing the full experimental process and the total time required to perform a genome edit -- including plasmid transformations, genome editing, and finally plasmid curing.

PLEASE NOTE As part of the EMBO Publications transparent editorial process initiative (see <http://www.nature.com/msb/journal/v6/n1/full/msb201072.html>), Molecular Systems Biology now publishes online a Review Process File with each accepted manuscript. Please be aware that in the

event of acceptance, your cover letter/point-by-point document will be included as part of this file, which will be available to the scientific community. Authors may opt out of the transparent process at any stage prior to publication (contact us at msb@embo.org). More information about this initiative is available in our Instructions to Authors.

If you feel you can satisfactorily deal with these points and those listed by the referees, you may wish to submit a revised version of your manuscript. Please attach a covering letter giving details of the way in which you have handled each of the points raised by the referees. A revised manuscript will be once again subject to review and you probably understand that we can give you no guarantee at this stage that the eventual outcome will be favorable.

In addition, I would like to acknowledge that this review process took somewhat longer than is usual at Molecular Systems Biology, due to some initial delays in finding reviewers during the December holiday season, and some delays in ultimately receiving the reviewers' reports.

REFeree REPORTS:

Reviewer #1:

Synopsis:

The authors combine two well-developed strategies, namely the Cre/lox recombinase system and mobile Group II intron system, both typically used for genome level recombination based genetic manipulations. They use previously developed algorithms for the predictions of intron target sites from *E. coli* and *L. lactis*. They also modify the flexibility of lox insert sequences to obtain improved efficiencies. This is a smart use of known methods and ideas to develop a potentially useful method for large-scale modifications in bacterial genomes. The ability to execute not only deletions and insertions but also inversions and exchanges is good, but should be considered an extension of known capabilities of the Cre/lox system. However, the ability to simultaneously remove or insert into multiple genes/ loci is particularly powerful in non-model organisms (where plasmid poration is possible).

Main concerns:

1) The manuscript is presented mainly as a methods development study. However, neither great breadth (numerous organisms) nor great depth (genome coverage) is demonstrated. The lac genes do serve as a good starting point having well defined impact on cell physiology when manipulated. However it is difficult to evaluate if the strategy will work efficiently for a wide distribution of genes/ operons/ regulons with respect to deletions and insertions. For instance, do different loci present different efficiencies? Does the size of the insert impact the efficiency of integration (or is there a relationship to delivery plasmid copy number/ time requirement)? As suggested by the authors, is it in fact possible to insert genes/ operons closer to origin stably and with high efficiency. While successful demonstration of each type of manipulation is presented nicely, there is not much experimentation that allowed me to assess the limits of this method.

2) A few aspects of the process may require additional clarification and if needed, experimentation. It appears that the efficiency of the process requires a balance between the time needed for successful interaction between the plasmid and chromosome (2days better than 1), but too much plasmid (low copy better than high) leads to undesirable Cre mediated swapping between plasmids. That an optimal balance is needed seems to also be suggested by the differing preference of these two parameters in the two strains. Is it worthwhile to modulate the levels of Cre protein (reduce) and plasmid copy number (increase) so as to maximize plasmid - chromosome interaction but minimize Cre mediated undesirable swapping? This may cut short the time required.

3) How does the copy number and time taken by the pET26b-DEBS1TEi plasmid compare with that of the pUC19 and pAC GFP containing plasmids? And was this also conducted in both *E. coli* strains (not clear from methods)?

4) The methods indicate that initial growth and maintenance of the strains required antibiotics, but subsequent steps for induction etc. did not use the selectable markers. Is this universally applicable,

irrespective of delivery plasmid copy numbers and size of insert (such as in the case of the 12Kb PKS operon containing pET26b-DEBS1TE plasmid? Further, was this also the case for *S. aureus*?

5) The author comment upon the need for flexibility in the structure of the inserts. Could they draw a more systematic conclusion about the structure - efficiency relationship of the lox-site inserts and insertion efficiency? For example 1L66, 1L71 and 1WL2 look similar but have different efficiency and even "preference" of introns.

6) This point is regarding how well this method will be used by others: Use of plasmid-borne genes/ pathways is a regular practice in strain engineering. Despite the need to use selection markers, the ease of testing numerous variations of genes/ pathways and their corresponding regulation via different control systems, make plasmid-based work irreplaceable in research areas such as metabolic engineering and synthetic biology. The strategy developed in this study may potentially present a way to enable genome level manipulation at the relative ease of plasmid-based methods. However, a systematic comparison of the Cre/Lox-targetron vs. a plasmid-based system may be needed to examine this. I would expect that the Cre/Lox-targetron system would have the clear advantage of being able to incorporate large pieces of DNA but the plasmid-based system would have the advantage of better range in copy numbers. However, are other factors equivalent (efficiency, stability, throughput, ease of use)? I am curious if the authors would like to take the time to explore this to a greater extent than they currently have.

Minor points:

1) It would be very helpful to have better description of the plasmids in supplementary Table 2. Description should contain copy number, ori, selection marker, promoter used, plasmid sizes and other pertinent details. It is hard for a reader to compile this information using sections of the main manuscript and other supplementary information and references. In some cases there is no published reference (provided) of the starting plasmid (e.g. pET26b-DEBS1TE).

2) In methods, provide generic name of materials or equipment used followed by the manufacturer in parentheses, rather than only a trademarked name. E.g. "96 well Nunc plate", Bio-Tek Power wave (p19)

3) Define acronyms and provide references for statistical methods used. E.g., ANOVA, Bartlett's test, Shapiro-Wilks normality test etc. (p20)

4) It may be useful to provide references for various growth media for *S. aureus*. E.g. TSA, TSB (p20)

5) The use of superlatives such as "incredibly" (see p13 ln 6) should be avoided.

6) Abbreviated names have been used (in several places) prior to first mention of the bacterium.

Reviewer #2 :

Enyeart et al combined two powerful tools (mobile group II intron-based "targetrons" and Cre-lox recombinase) for engineering genomes and plasmids in broad range of cells in order to create a generalized system for direct manipulation of bacterial chromosomes. Importantly, they show their system works efficiently in two vastly different bacteria (*E. coli* K-12 and a *Staphylococcus aureus* strain). Since both targetrons and the Cre-lox recombinase has previously been shown to work in diverse bacteria (as they document), it is reasonable to concur their system will be broadly applicable.

Primary "novel feature" is their data concerning how to design targetrons that operate efficiently. Joining these systems to work together and demonstrating their utility, especially in bacteria for which there exists a paucity of genome manipulation tools is the major strength of this manuscript. Supplementary Figure 1 is too general and Supplementary Table 1 is difficult to follow without reference to other manuscripts. It would be good if others created a revised Supplementary Figure 1 in which they use a single example from Supplementary Table 1 and show precisely where different sequences are located. This would help guide the more interested reader to adopt the methodology.

One error exists, which should be corrected throughout. pACYC184 is NOT a low-copy plasmid. Indeed, its copy number is close to 15 to 25 copies per cell. Low-copy generally refers to 1 to a few copies per cell. Low-copy should be changed to medium-copy throughout. Bottom of page 7 mentions HMS174 (and BL21), which I failed to find elsewhere. I assumed HMS174 is a derivative of MG1655, which is mentioned in Supplementary material

Reviewer #3:

This paper by Enyeart et al describes a new method for making genome rearrangements based on the catalytic activity of engineered group II introns, called targetrons, and the site-specific recombination system, Cre-Lox. Using their method, the authors were able to "edit" the genome of *E. coli* in different ways. These included plasmid-to-chromosome recombination, chromosomal deletions of ~120 Kb, along with translocations and inversions involving even larger segments of DNA. Some inversions proved to be unstable, but this potential drawback of the method was solved by using targetrons comprised of non-homologous group II introns. Portability of this method was also shown. For this experiment, the authors made a targeted 15-Kb deletion in the genome of *S. aureus*, a Gram-positive anaerobic bacterium. The authors argue that their "genome editing" technique holds advantages over recombineering based on three factors: portability, high-efficiency, and the unique capacity to bring about both inversions and translocations.

Although this methods paper describes useful advances in bacterial genome manipulation, the presentation of the work is inadequate and uneven:

1. The unevenness of the paper is reflected in extensive details relating to the statistical methods, but comparatively little on the molecular biology. It is information of the latter category that is of greatest interest to potential users of the technique. We therefore recommend fixing this imbalance, moving the detailed statistics to supplementary material and providing better descriptions of the genetic manipulations.

2. There is no really informative figure of how the group II intron delivers the lox sites. Fig. 3A is not sufficiently informative, and there is no sense of scale. In general, the figures require a lot of work.

3. The details and description as well as the visual representation of results are insufficient and often confusing. For example, a few parts of the manuscript entitled 'Deletions'; 'Inversions'; 'One-step cut-and- paste' and corresponding Figures 5-7 do not complement each other well. Figures seem to be too casually referenced in the main text, while text itself provides some obscure information which is not related to the data. Much of what is in the figure legends should be in main text with more relevant details.

4. Figures carry discrepancies, especially in gel pictures. Bands appear in some lanes where their presence is surprising or the sizes do not correspond to the predicted. Figures with gels should be clearer, with better schematic representation for localization of the primer pairs (especially Figure 8). Bands should be labeled.

5. The authors describe their technique as a "one-step" cut-and-paste; however it seems that multiple steps are required to effect genome editing. The user must introduce a plasmid to express the engineered targetron, a plasmid to express Cre, then a third plasmid harboring the additional DNA segment, at least in the case of recombination. After recombination has occurred, the user would presumably want to cure the cell of these plasmids, which although tedious, might be sped up by introducing counter-selectable markers on the plasmids.

6. The authors demonstrate the portability of their method by making a deletion in the chromosome of *S. aureus*. Unlike *E. coli*, *S. aureus* is Gram-positive. A more rigorous test of portability might be in an organism like mycobacteria, whose high GC content might confound the design of targetrons.

7. Some simple explanations of the following would be in order:

- a. It's stated that LtrB.LacZ.635s and Ec15.LacZ.912s were used (p6). What do the numbers refer

- to? Why, then, on p7 is it stated the EcI5.LacZ1806s intron is used to deliver the lox sites?
- b. p6: An argument is made for intron efficiency based on flexible versus rigid structures; how was flexibility/rigidity evaluated?
- c. p7: "Two introns modified *E. coli* strains, HMS174 and BL21, were used and were plated at one, two, and three days after transformation." How were they modified?
- d. p8: Move "A multifunctional analysis...." out of MS and into Supplementary materials. Should "interactions" in that paragraph be "relationships"?
- e. p8: Might the better performance of the low-copy number vector be related to toxicity effects?
- f. p9: "Insertion of the entire operon.." into what? Where? Also, the lacY effect is unclear.

Second Submission

31 May 2013

Please find attached a revision of "Generalized bacterial genome editing using mobile group II introns and Cre-lox" (Manuscript Number: MSB-11-3454) by Enyeart and coworkers. We thank the reviewers for their helpful comments regarding our work. We recognize that more than three months has passed since we received the reviewer's comments, but we feel the work continues to be relevant and thus respectfully request that you consider this revised manuscript for publication in *Molecular Systems Biology*.

We have taken the intervening time to add a great deal of new data to the manuscript, including comparisons of the efficiency of recombination-mediated cassette exchange (RMCE) at different loci in the *E. coli* genome, and the intron-mediated delivery of lox sites to the genomes of two new species, *Bacillus subtilis* and *Shewanella oneidensis* (the first report of successful targetron use in these species). To assist other labs with these techniques, we have made the intron-retargeting algorithm available at <http://www.targettrons.com>, and have developed plasmid vectors suitable for delivering DNA fragments to genomic loci via Cre-lox. Responses to specific questions and concerns raised by the reviewers can be found below.

Reviewer #1

The manuscript is presented mainly as a methods development study. However, neither great breadth (numerous organisms) nor great depth (genome coverage) is demonstrated.

As noted, we have increased the number of organisms from two to four (see the red text in the section on "Genome engineering in diverse bacteria," pp. 16-18, and **Supplementary Figures 4 and 5**). The additional organisms included one Gram positive and one Gram negative bacteria, and both were bacteria in which targettrons had not previously been demonstrated to be functional. This work further builds on the already known broad range of organisms in which both targettrons and Cre/lox have been shown to function, strengthening our hypothesis that our genome engineering methods should work in many different bacteria. As for genome coverage, we have increased the number of loci examined for RMCE efficiency in *E. coli* from one to three (see the paragraph in red starting on p. 8 and ending on p. 9, and **Figure 5C**), and have demonstrated deletions, inversions, and transversions involving a number of different loci that are widely separated in the *E. coli* genome.

For instance, do different loci present different efficiencies?

In the case of RMCE, we have taken the time to address this issue using the previously demonstrated method of including a T7 promoter in the intron just before the inserted lox sites. This allows us to insert the promoter directly into the genome prior to inserting a GFP gene via Cre/lox. We focused on loci that could easily be screened visually (the *galK* and *malT* genes, which can be screened for functionality on MacConkey agar), but as the set of *lacZ*, *galK*, and *malT* genes are widely spaced in the *E. coli* genome (see the updated version of **Figure 3** (formerly **Figure 2**)) more general coverage can be inferred. The new data have been merged into **Figure 5** (formerly **Figure 4**) and are discussed in the red paragraph starting on p. 8 and ending on p. 9. While there were some positional effects, we once again observed a higher efficiency of insertion after two days of incubation, confirming previous results. In all cases the efficiency was high enough that insertions could be easily identified via a small number of colony PCRs. Similarly, there were no significant differences in the efficiencies of deletions or inversions at different loci: all the deletions and inversions that were tolerated by the cells occurred at essentially 100%, and taken together represent a broad sampling of different genome regions.

Does the size of the insert impact the efficiency of integration (or is there a relationship to delivery plasmid copy number/ time requirement)?

The size of the insert does not seem to be a significant factor. Both the insertion of GFP (a cassette of approximately 1 kb) and the polyketide-synthase operon (12 kb in length) occurred at close to 100% after a three-day incubation, based on colony PCR. We have added the following text to the Discussion section (see p. 21): "The use of the Cre/lox system also allows large pieces of foreign DNA to be integrated into genomes at high efficiency. An initial recombination occurs between a lox site on the plasmid and a lox site in the genome, serving to integrate the entire plasmid into the genome. We found no evidence for a difference in efficiency between inserting 1 kb and 13 kb into the *E. coli* genome via RMCE. Given the high efficiency observed during the construction of large deletions and inversions, the limiting factor in RMCE would thus seem to be the initial encounter between the plasmid and the genome. A second recombination event that occurs between the other two lox sites removes the plasmid sequence. The speed and efficiency of this second recombination event is rapid and essentially 100% efficient, similar to the other intragenomic recombinations we report."

Is it worthwhile to modulate the levels of Cre protein (reduce) and plasmid copy number (increase) so as to maximize plasmid - chromosome interaction but minimize Cre mediated undesirable swapping?

The levels of the Cre protein are already fairly low, as it is expressed from a vector (pQL269, with the pSC101 origin) with a relatively low copy number of about 5 per cell (see p. 1.8 of the 2001 edition of *Molecular Cloning* by Sambrook and Russell) under the control of an uninduced lac promoter and in the presence of LacI. Moreover, the efficiencies of insertions seem to rise on the second day regardless of plasmid copy number, strain, or locus. Thus, there is no reason to believe that decreasing the amount of Cre further will significantly impact efficiency of insertion on the first day, and even were we to obtain this result, it would not significantly improve the method.

How does the copy number and time taken by the pET26b-DEBSITEi plasmid compare with that of the pUC19 and pAC GFP containing plasmids? And was this also conducted in both E. coli strains (not clear from methods)?

We have added the following to the text to clarify the issue of copy number (see p. 9 lines 7 to 9): "The pET vectors are built on the pBR322 backbone (Rosenberg et al, 1987), which is similar in copy number to the pACYC backbone used for the pACD plasmids (Green & Sambrook, 2012)." As noted in the text (p. 9, line 5), the DEBSITE insertion was performed in *E. coli* K207-3, which is a strain specifically engineered for the expression of polyketide synthases.

The methods indicate that initial growth and maintenance of the strains required antibiotics, but subsequent steps for induction etc. did not use the selectable markers. Is this universally applicable, irrespective of delivery plasmid copy numbers and size of insert (such as in the case of the 12Kb PKS operon containing pET26b-DEBSITE plasmid? Further, was this also the case for S. aureus?

Simply put, yes. Selectable markers are required for maintaining the various plasmids in the cells, but not for ensuring or maintaining RMCE or integration of targetrons. This statement holds for all types of recombinations in all species reported in the manuscript. While targetrons have been engineered that allow selection for insertion, these were not used in the present study. We have added the following statement to the Discussion section to clarify this point (see p. 19, lines 11 and 12): "while the use of selectable markers is required for temporary plasmid maintenance, markers need not be used for selecting or maintaining genomic modifications."

The author comment upon the need for flexibility in the structure of the inserts. Could they draw a more systematic conclusion about the structure - efficiency relationship of the lox-site inserts and insertion efficiency? For example 1L66, 1L71 and 1WL2 look similar but have different efficiency and even "preference" of introns.

The details of RNA folding are complex and incompletely understood, especially for molecules as large and intricate as targetrons. Fine details in the differences in folding and mechanism between the two types of introns may have differentially impacted what sequences and structures could be functionally inserted, but determining the underlying mechanistic differences due to folding would be close to impossible and certainly beyond the scope of the current work.

We also note that many of the observed differences were not statistically significant. For instance, the *P*-values for the pair-wise differences in the efficiency between the 1L66, 1L71, and 1WL2 inserts in the LtrB intron were all greater than 0.95, indicating that those differences that were observed could easily have arisen by chance. In EcI5 the 1WL2 insert was significantly different from 1L66 and 1L71, but 1L66 and 1L71 were not significantly different from each other (*P*-value = 0.87).

Use of plasmid-borne genes/ pathways is a regular practice in strain engineering. Despite the need to use selection markers, the ease of testing numerous variations of genes/ pathways and their corresponding regulation via different control systems, make plasmid-based work irreplaceable in research areas such as metabolic engineering and synthetic biology. The strategy developed in this study may potentially present a way to enable genome level manipulation at the relative ease of plasmid-based methods. However, a systematic comparison of the Cre/Lox-targetron vs. a plasmid-based system may be needed to examine this. I would expect that the Cre/Lox-targetron system would have the clear advantage of being able to incorporate large pieces of DNA but the plasmid-based system would have the advantage of better range in copy numbers. However, are other factors equivalent (efficiency, stability, throughput, ease of use)? I am curious if the authors would like to take the time to explore this to a greater extent than they currently have.

In short, plasmid-based approaches are insufficient when the structure and function of the genome is the object of a study. Moreover, working with genomes allows one to deal with much larger chunks of DNA than would be feasible using plasmids or even BACs. Finally, for toxic proteins, expression from the genome may allow lower levels of expression than can be obtained on plasmids (for an example, see Gescher et al. Mol. Microbiol. 68:706-719 (2008), in which the experimenters "chose integration of the *cymA* and *napC* genes into the genome of *E. coli* as previous experiments with expression from plasmids displayed a clear toxic effect due to a higher gene dosage"). We have added a mention of these issues to the discussion on p. 18, line 21, p. 19, line 1: "[The genome as the unit of engineering] allows much larger swaths of DNA to be manipulated than is possible with plasmid-based methods and enhances our ability to study the structure and function of genomes."

It would be very helpful to have better description of the plasmids in supplementary Table 2. Description should contain copy number, ori, selection marker, promoter used, plasmid sizes and other pertinent details. It is hard for a reader to compile this information using sections of the main manuscript and other supplementary information and references. In some cases there is no published reference (provided) of the starting plasmid (e.g. pET26b-DEBS1TE).

We have added such information for all the vectors used in this study (see **Supplementary Table 3**, previously **Supplementary Table 2**). A description of the construction of the pET26B-DEBS1TE plasmid has been added to the Supplement (see p. 5 in the Supplement, the red text in the paragraph titled ("Generating the polyketide synthase insertion donor plasmid").

In methods, provide generic name of materials or equipment used followed by the manufacturer in parentheses, rather than only a trademarked name.

We have done so, specifically for the 96-well plates and plate reader that were used for the doubling time measurements (see p. 27, lines 13 and 15).

Define acronyms and provide references for statistical methods used.

We have done so. Specifically, acronyms have been clarified for recombination mediated cassette exchange/RMCE (p. 7, line 7), polymerase chain reaction/PCR (p. 9, line 2), Luria-Bertani/LB (p. 25, line 12), tryptic soy broth/TSB (p. 26, line 2), tryptic soy agar/TSA (p. 26, line 13), open reading frame/ORF (p. 26, line 18). Detailed statistical discussions of statistical methods and results have

now been moved the supplement, but the first paragraph of the newly created **Supplementary Text** defines the ANOVA (analysis of variance) acronym and cites references for analysis of variance, Bartlett's test for equality of variances, the Shapiro-Wilk test, and the Tukey method of multiple comparisons.

It may be useful to provide references for various growth media for S. aureus. E.g. TSA, TSB

We have amended these mentions to state the full names (tryptic soy agar and tryptic soy broth) of the media (p. 26, line 13, and p. 26, line 2) and assume this will resolve any uncertainties.

The use of superlatives such as "incredibly" (see p13 ln 6) should be avoided.

We have done our best to remove such hyperbole throughout.

Abbreviated names have been used (in several places) prior to first mention of the bacterium.

We believe that we have corrected all such problems. The first mention of *Escherichia coli* (p. 3 line 10) is unabbreviated, as are the newly added first mentions of the other bacteria used in the present study (p. 5 lines 8-9). The first mention of *L. lactis* was corrected to *Lactococcus lactis* (p. 3, line 19).

Reviewer #2

Supplementary Figure 1 is too general and Supplementary Table 1 is difficult to follow without reference to other manuscripts. It would be good if others created a revised Supplementary Figure 1 in which they use a single example from Supplementary Table 1 and show precisely where different sequences are located. This would help guide the more interested reader to adopt the methodology.

We have moved the former **Supplementary Figure 1** into the main text as **Figure 1** and have further modified it to show additional structural and mechanistic detail. In addition, the former **Figure 8** has been moved to the supplement as **Supplementary Figure 3**.

One error exists, which should be corrected throughout. pACYC184 is NOT a low-copy plasmid. Indeed, its copy number is close to 15 to 25 copies per cell. Low-copy generally refers to 1 to a few copies per cell. Low-copy should be changed to medium-copy throughout.

While some sources, such as Qiagen (<http://www.qiagen.com/Knowledge-and-Support/FAQ/?ID=1f42840e-fbd7-4734-b0cd-e17372a9e5a4>) do describe pACYC184 as a "low-copy" vector, the point is taken that the language used could be misleading. We have changed **Figure 5** (formerly **Figure 4**) to show numerical copy numbers instead of verbal descriptions of "high" or "low" and have similarly amended the main text (p. 7 lines 19-20; p. 8 lines 13 and 15).

Bottom of page 7 mentions HMS174 (and BL21), which I failed to find elsewhere. I assumed HMS174 is a derivative of MG1655, which is mentioned in Supplementary material

HMS174 is a derivative of W3110 (see Campbell et al. Genetic recombination and complementation between bacteriophage T7 and cloned fragments of T7 DNA. *PNAS* **75**:2276-2280 (1978)), which is a K-12 strain closely related to but not a direct descendant of MG1655 (see Hayashi et al. Highly accurate genome sequences of *Escherichia coli* K-12 strains MG1655 and W3110. *Mol Sys Biol* **2**:2006.0007). We have made additions to the text to clarify this (p. 7, line 22 to p. 8, line 1).

Reviewer #3

The unevenness of the paper is reflected in extensive details relating to the statistical methods, but comparatively little on the molecular biology. It is information of the latter category that is of greatest interest to potential users of the technique. We therefore recommend fixing this imbalance, moving the detailed statistics to supplementary material and providing better descriptions of the genetic manipulations.

We have moved the detailed statistical discussions to the supplement and replaced them with simple explanations of the results of the statistical analyses. Specifically, we replaced the statistical discussion of **Figure 2** (formerly **Figure 1**) with: "Statistical analyses (see **Supplementary Text**) confirmed that the inserts fall into two classes: one of wild-type efficiency, and one of impaired efficiency" (see p. 6, lines 14-15).

We replaced the statistical discussion of **Figure 5B** (formerly **Figure 4B**) with: "A statistical analysis of these results (see **Supplementary Text**) indicated that time and delivery-plasmid copy number were significant factors, but that strain type was not. However, the interaction of strain type with the other factors was significant. Furthermore, significant differences were found between the first day and subsequent days, but not between the second and third days" (see p. 8, lines 4-8).

We replaced the statistical discussion of Table 2 with: "A statistical analysis of the doubling times indicated that the strains fall broadly into two groups, one group having approximately wild-type growth, and the other group having impaired growth (see **Supplementary Text**)" (see p. 16, lines 15-17).

We have also put the statistical details of the new analysis of the effect of genomic location on RMCE (see **Figure 5C** and p. 8, line 18 to p. 9, line 2) into the **Supplementary Text**.

We have also done our best to improve the descriptions of the genetic manipulations (revisions on pp. 10-16, further discussed below).

There is no really informative figure of how the group II intron delivers the lox sites. In general, the figures require a lot of work.

We originally hoped that **Supplementary Figure 1** would explain the mechanism of intron integration. We have now moved this Figure to the main text, as **Figure 1**. To further clarify mechanistic details, we have also added more sections to the figure to show intron structure and the details of the base-pairing interactions that facilitate integration at the target site.

Fig.3A is not sufficiently informative, and there is no sense of scale.

Figure 3A (now **Figure 4A**) is now supported by much more context. The new version of **Figure 1** provides much more detail on intron mechanism. **Figure 3** (formerly **Figure 2**) provides a complementary sense of the genomic scale of the recombination events. **Figure 4** (formerly **Figure 3**) then shows how Cre and lox sites interact during the execution of the rearrangements. In this case, though, the scale involved will vary considerably depending on the specific rearrangement under consideration (or, in the case of **Figure 4A** (formerly **Figure 3A**), on the size of the fragment being delivered). Thus, in this new context, **Figure 4** can be seen to represent an integration of mechanisms, from the small lox sites to the large cassettes being delivered.

The details and description as well as the visual representation of results are insufficient and often confusing. For example, a few parts of the manuscript entitled 'Deletions'; 'Inversions'; 'One-step cut-and-paste' and corresponding Figures 5-7 do not complement each other well. Figures seem to be too casually referenced in the main text, while text itself provides some obscure information which is not related to the data. Much of what is in the figure legends should be in main text with more relevant details.

We have reduced the amount of text in the figure legends and significantly expanded the analysis of the gels in the main text, the details of which follow:

On pp. 10-11 we have added a much more detailed description of how the deletions were verified:

"Three PCR amplifications can be used to characterize each deletion: two amplicons that bridge the genomic sites at which lox-carrying targetrons are inserted, and one amplicon that bridges the expected deletion between those sites. The first two PCRs testing for insertion are expected to give relatively small bands (several hundred base pairs) when performed on the wild-type strain and larger bands (about 1 kb larger) when performed on a strain harboring targetrons at the expected sites. Upon successful deletion of the intervening region, all of these bands should no longer be generated. Instead, the PCR that bridges the two sites should yield a new band of a predicted size. For example, the PCR product derived from a deletion between the lacZ and A loci should be 1236 bp in length, and the sequence should match the genome sequence upstream of the A insertion site, followed by the 5' sequence of the LtrB.A intron up to the lox72 site (created by recombination), then the 3' sequence of the EcI5.LacZ.1806s intron, and finally the sequence of the lacZ gene

downstream of the intron insertion site. Doing all three PCR amplifications on all three strains (wild-type, wild-type harboring insertions, and recombined (induced with Cre)) provided clear diagnostic descriptions of the recombination events. When artifact bands near the sizes of expected bands were observed, these were further analyzed by sequencing to ensure that they did not represent an off-target rearrangement. The predicted sizes of all of the expected amplicons of the present work (as shown in the gels of **Figures 6 through 8** and **Supplementary Figures 3 through 5**) are listed in **Supplementary Table 2**."

More detailed descriptions of the gels in **Figure 6** (formerly **Figure 5**) have been added to pp. 11-12:

"In particular, **Figure 6B** shows the three PCR amplifications performed on the three successive, engineered strains that ultimately resulted in a deletion of 121 kb between the *A* and *lacZ* loci. The deletion-bridging amplicon (Au/Ld-I in **Fig. 6**) from the strain that was finally exposed to the Cre protein (*E. coli* MG1655 E1) was sequenced and found to conform to expectations. All the bands seen in this gel were predicted by the analysis shown in **Figure 6A**.

"As deletions are added, verification becomes more complex but is performed in exactly the same manner. **Figure 6C** shows the PCR amplifications for verifying the sequential double-deletion strain (*E. coli* MG1655 E6) that harbors deletions of both the *A-lacZ* and *D-E* regions. The first triplet of reactions (Au/Ld) verifies the continued presence of the *A-lacZ* deletion. The second triplet (Eu/Dd) should only yield a band if the *D-E* region has been successfully deleted. The band is small because the recombination results in an inverted repeat of intron sequences that is subsequently removed by homologous recombination. Sequencing results confirmed this interpretation. The other triplet PCR amplifications in **Figure 6C** confirm the original insertion sites of the introns. In this instance, as opposed to the reactions shown in **Figure 6B**, there were additional bands observed following PCR, as might be expected given the deletion of templates that some of the primers were originally designed to amplify. These bands, in particular the Eu/Dd-I, Au/Ad-U, and Au/Ad-I bands, were sequenced and found to match genomic sequences from unrelated regions, and not to off-target rearrangements.

"Similarly, **Figure 6D** shows the PCR amplifications for verifying the sequential triple-deletion strain (*E. coli* MG1655 E10) that contains a deletion of the *B-C* region in addition to the *A-lacZ* and *D-E* regions. The first two triplet PCRs verify the continued presence of the *A-lacZ* and *D-E* deletions. The third triplet (Bu/Cd) verifies that the *B-C* deletion is not seen until the *lox* sites have been both introduced and recombined. As before, genomic PCR artifacts are sometimes seen in the absence of the expected amplicon; the band in the Bu/Cd-U lane was sequenced and again found to be from an unrelated genomic region. The Bu/Cd-U amplicon that confirms the deletion also represents the formation and removal of an inverted repeat, which was confirmed by sequencing. The last two triplets verify the insertion of introns at the *B* and *C* loci that are not seen once Cre is added. The simultaneous triple deletion strain (E9) and a strain containing a single deletion of the *D-E* region (E11) were verified in the same manner."

A clarification of the method used to assess inversions was added to p. 13, lines 7-10: "The same methods used to detect deletions can be used to detect inversions, except that four different PCRs are used to verify an inversion: two amplicons bridging the insertions sites and two amplicons bridging the new ends of the inversion (see **Fig. 7A**)."

Much more detailed descriptions of the gels in **Figure 7** (formerly **Figure 6**) have been added to pp. 14-15:

"**Figure 7B** shows the PCRs used to verify an inversion between the *A* and *lacZ* loci, as depicted in **Figure 7A**. **Figure 7C** shows an analogous set of PCRs for verifying an inversion between the *D* and *E* loci. Since these introns are present in opposite orientations upon integration, inversions can occur via homologous recombination, and this inversion is in fact detected at low levels prior to the addition of Cre. Similarly, the uninverted (reverted) state can still be detected after the addition of Cre. These bands were confirmed by sequencing. In those strains where inversion had occurred in the absence of Cre, unrecombined *lox* sites were found, while in those strains where inversions back to the wild-type state had apparently occurred after the induction of Cre, recombined *lox* sites were found, consistent with homologous recombination between the introns rather than catalyzed recombination between the *lox* sites. No artifacts were seen in these instances, consistent with the presence of a template that could be amplified by the primers, even if that template had arisen by homologous rather than Cre-mediated recombination.

"**Figure 7D** demonstrates the difference between using homologous introns (where both introns are of the EcI5 type) versus non-homologous introns (one EcI5, one LtrB) for delivering *lox* sites to create substantially identical inversions. When non-homologous introns are used, as in the E5 strain, the inversion is only detected (via the Lu₀/Ed-I and Eu/Ld₀-I bands, which are of the expected

size and were confirmed by sequencing) after adding Cre, and reversions back to the original state via homologous recombination were not detected (i.e., the uppermost Eu/Ed-U and Lu₀/Ld₀-U bands, that correspond to the expected sizes for Cre-mediated inversions and that were also confirmed by sequencing, were not seen in the corresponding "I" lanes). However, when homologous introns were used, as in the E2 strain, PCR products that should only have been seen upon inversion were also seen in the absence of Cre, and further the uninverted state could still be detected after the addition of Cre, consistent with homologous recombination between introns. As before, the bands of the sizes expected for recombination events, whether due to Cre-*lox* or homologous recombination, were confirmed by sequencing."

A clarification of the methodology for assessing cut-and-pastes was added to p. 15: "Six different PCR amplifications were used to validate a stable cut-and-paste reaction, as shown in **Figure 8**: three amplicons bridging the three intron integration sites (the left set of three triplets in **Figures 8A** and **B**), one amplicon bridging the site of the "cut" (the fourth set of triplets in **Figs. 8A** and **B**), and two amplicons bridging the boundaries of the "paste" region (the fifth and sixth set of triplets in **Figs. 8A** and **B**)."

Finally, a more detailed analysis of the gels in **Figure 8** (formerly **Figure 7**) was added to pp. 15-16: "In the case shown in **Figure 8A**, intron homology allows inversions to occur back and forth between the *lacZ* and *E* loci (seen by the continued presence of the unrecombined Lu/Ld band in the "I" lane and the recombined Lu/Ed band in the "U" lane), but the complete cut-and-paste (as judged by the presence of all three of the Au/Ld-I, Lu/Ed-I, and Eu/Ad-I bands, which are of the expected sizes and were further verified by sequencing) was only seen upon addition of the Cre protein. **Figure 8B** shows a similar case where the orientation of the *lox* site at the target (*E*) locus is reversed with respect to the case shown in **Figure 8A**. In this case the bands in the Au/Ld-I, Eu/Lu-I, and Ed/Ad-I lanes, which are of the expected size and were verified by sequencing, again demonstrate that the expected rearrangement was obtained. However, since the insertion at *E* is in the opposite orientation inversions and reversions resulting from intron homology were avoided."

Figures carry discrepancies, especially in gel pictures. Bands appear in some lanes where their presence is surprising or the sizes do not correspond to the predicted.

There was a case in the former **Figure 5** (now **Figure 6**) where the band sizes stated did not correspond exactly to the sizes of the bands in the gel. This was due to erroneously using the sizes expected for a recombination between the *A* and *lacZ.635* loci, instead of the *A* and *lacZ.1806* loci, which was the recombination actually executed in this case. So, while the data was consistent, the labeling was not. We thank the reviewer for pointing out this error, which has been corrected.

All bands in the figures fall into one of four classes: (1) bands of the expected size; (2) bands of smaller than expected size resulting from deletion of inverted repeats via homologous recombination; (3) bands of expected size in unexpected lanes, which typically resulted either from (a) homologous recombination or (b) amplification of unrelated genomic regions; and (4) dim bands of a variety of sizes, assumed to be artifacts. Bands in classes (1), (2), and (3) were all sequenced; bands in class (4) were typically not. We have attempted to make clear which bands correspond to which classes in the text quoted above from the subsections under Results on deletions, inversions, and one-step cut-and-pastes on pages 10 to 16. Generally speaking we are confident that the control experiments and the sequencing results (which are now extensively reported throughout the text) fully demonstrate the results we claim.

Besides expanding the analysis of the gels in the main text, we have also added a new Table (**Supplementary Table 2**) that lists the expected sizes of all the PCR amplicons appearing in the gels of **Figures 6** through **8** and **Supplementary Figures 3** through **5**. All of these expected sizes are based on the annealing sites of the relevant primers (listed in **Supplementary Table 5**), the knowledge that a *lox*-carrying LtrB intron is 955 bp and a *lox*-carrying EcI5 intron is 919 bp, and the intron integration sites (listed in **Supplementary Table 1**).

Figures with gels should be clearer, with better schematic representation for localization of the primer pairs (especially Figure 8).

We have added a schematic representation of the localization of primers pairs to the former **Figure 8** (now **Supplementary Figure 3B**). We hope that this modification combined with the modifications to the text and other Figures and Tables mentioned above will now rectify this issue.

Bands should be labeled.

Labeling all of the bands on the gels would have cluttered the figures to the point of unreadability. Instead, we have accounted for the sizes and sequences of all significant bands on all gels, as detailed in **Supplementary Table 2**. In so doing, we have gone well beyond merely labeling the bands to provide near comprehensive explanations for the provenance of these bands.

The authors describe their technique as a "one-step" cut-and-paste; however it seems that multiple steps are required to effect genome editing. The user must introduce a plasmid to express the engineered targetron, a plasmid to express Cre, then a third plasmid harboring the additional DNA segment, at least in the case of recombination.

The cut-and-paste procedure is called "one-step" because once the *lox* sites are in place, the transversion can be executed in one step: by adding Cre. Mechanistically it is, in fact, one step. We believe this is the important point, since in almost any other method the region to be transferred might easily be lost without purification or amplification; i.e., one would need to first transfer the region of interest onto a plasmid, BAC, or other piece of mobile DNA, and then transfer it from there to the new location in the genome. By artfully arranging the recombinase sites, the time-consuming intermediate shuttle step can be skipped. We have attempted to clarify this issue in the text by adding the following text (see p. 15 lines 10-12): "We use the term "one-step" because the designated region moves directly from one part of the genome to another upon adding Cre, without the requirement for a stable intermediate, such as a plasmid, to act as a shuttle."

After recombination has occurred, the user would presumably want to cure the cell of these plasmids, which although tedious, might be sped up by introducing counter-selectable markers on the plasmids.

The Cre plasmids all have temperature-sensitive origins of replication, and the counter-selectable marker *SacB* is included on the RMCE delivery vectors, making curing of the plasmids a simple matter. As for the intron plasmids, we found in all systems except *Shewanella* that a large fraction of colonies with intron insertions had lost the intron plasmid (as judged by loss of antibiotic resistance after plating intron-induction cultures on non-selective media). Interestingly, *Shewanella* is likely dependent on the continued presence of the intron-encoded protein (expressed from the intron-containing plasmid) for splicing of the inserted targetron out of the rRNA. We agree that this point should be made in the main text, and have thus added the following text to p. 22, line 21, to p. 23, line 5: "Removing the genome-modifying plasmids was also simple. Except in the case of *S. oneidensis*, which required the continued presence of the intron-encoded protein to allow the intron to splice out from the rRNA, a significant fraction (at least 1/3) of colonies were found to have lost the intron-expressing plasmid after the induction process. The Cre-expressing plasmids employed all contained temperature-sensitive origins of replication, and the delivery plasmids for RMCE had the *sacB* gene for counter-selection on sucrose, allowing these plasmids to be easily removed, as well."

We have also added a full description of the time involved in executing the method to pp. 21-22: "Targetron genomic engineering technology can be readily practiced by almost any lab. The algorithm for retargeting the targetrons is available online. The targeting sites in the intron can be changed via restriction cloning of a short fragment of DNA that can be created via two PCR reactions or synthesized in its entirety (see **Material and Methods**), followed by the typical time required for ligation, transformation, and sequence validation. Retargeting and the addition of *lox* sites can be performed for multiple introns in parallel. Following electroporation into the target strain, intron induction requires only one day, and plated induction colonies grow after one day. The method is similar in complexity to *lox*-site placements with lambda Red, but is an improvement on recombineering in that no selectable markers are required and it can be used in strains where lambda Red performs poorly. Similarly, Cre-mediated recombination requires one day for electroporation of the Cre-expressing plasmid (and, for RMCE insertions, the delivery plasmid), and one to two days for the cells to grow and for recombination to occur. Though we used plasmids (delivered by electroporation or conjugation) to deliver targetrons in the present study, phage, direct electroporation or other methods could potentially be used, as well."

The authors demonstrate the portability of their method by making a deletion in the chromosome of S. aureus. Unlike E. coli, S. aureus is Gram-positive. A more rigorous test of portability might be in an organism like mycobacteria, whose high GC content might confound the design of targetrons.

We have demonstrated the method in more organisms in the revised manuscript. While executing the method in a mycobacterium would indeed be useful and perhaps more impressive as a demonstration, the slow growth and other difficulties inherent in working with mycobacteria led to us to focus on demonstrating the method in a larger range of more typical workhorse strains for biotechnology.

It's stated that LtrB.LacZ.635s and Ec15.LacZ.912s were used (p6). What do the numbers refer to? Why, then, on p7 is it stated the Ec15.LacZ1806s intron is used to deliver the lox sites?

The numbers refer to the nucleotide position within the gene at which the intron inserts. We have added a clarification of this point to the text (see p. 6, lines 5-7): "the numbers 635 and 912 indicate the position in the *lacZ* gene at which the introns insert, and "s" (as opposed to "a") indicates that the introns insert into the sense strand of the gene." The 1806s intron is extremely efficient (97%) and is thus useful for general applications of the method. However, for testing what factors serve to reduce or increase intron efficiency, we worried that the 1806s intron might be so efficient that impediments to function might not reduce its mobility by significant margins. Thus we chose to use the lower-efficiency 912s (Ec15) and 635s (LtrB) introns for the experiments concerning the effect of inserts on introns efficiency (**Fig. 2**, formerly **Fig. 1**), and have added an explanation to that effect on p. 7, lines 8-10.

An argument is made for intron efficiency based on flexible versus rigid structures; how was flexibility/rigidity evaluated?

Flexibility was evaluated primarily on the basis of the presence or absence of a non-base-pairing region at the base of the hairpin structures, and secondarily on the size of these non-base-pairing regions. We have made this clarification in the text (see p. 6, lines 11-13).

"Two intron-modified E. coli strains, HMS174 and BL21, were used and were plated at one, two, and three days after transformation." How were they modified?

"Intron-modified" means the strains were modified by having introns inserted into their genomes. We have changed the text quoted to now read, "Two *E. coli* strains, HMS174 (a K-12 strain related to MG1655), and BL21 (a B strain), which contained intron-mediated insertions of *lox* sites..." (see p. 8, line 1).

Move "A multifactorial analysis...." out of MS and into Supplementary materials. Should "interactions" in that paragraph be "relationships"?

As discussed above, we have moved detailed statistical discussions into the **Supplementary Text**. "Interaction" is indeed the proper term for this type of statistical relationship.

Might the better performance of the low-copy number vector be related to toxicity effects?

Perhaps, but we do not think there is a relationship between Cre functionality and high-copy plasmid toxicity, since the amount of Cre protein (which in all RMCE experiments is expressed from a plasmid present at about 5 copies per cell) is likely not changing significantly. We have therefore chosen not to speculate here.

"Insertion of the entire operon.." into what? Where?

Into the *lacZ* gene of *E. coli* K207-3, as stated in the beginning of the paragraph cited. We have amended this sentence to make this explicit (see p. 9, line 10).

Also, the lacY effect is unclear.

As discussed in the reference provided (Mehdi et al., 1971), knocking out *lacY* decreases sensitivity to IPTG, presumably because less of it gets into the cell without the transporter. This reduces the likelihood of toxicity effects due to overexpression upon the addition of IPTG, and allows more fine-tuning of expression, since adding more IPTG will increase expression to a smaller degree. We

have added a clarification to the text that the effect of knocking out lacY is "due to a smaller amount [of IPTG] being transported into the cell" (see p. 9, line 14).

Overall, we have thoroughly answered the reviewers' concerns, have provided a great deal of additional data and analysis, and thus believe that this manuscript is now eminently suitable for publication in *Molecular Systems Biology*. Though some time has passed since the original submission, the work remains extremely relevant (perhaps even more so, now that it is clear that we continue to have clear advantages over other genome editing methods).

We look forward to working with you on the publication of this and other manuscripts, and remain

2nd Editorial Decision

07 July 2013

Thank you again for submitting your work to Molecular Systems Biology. We have now heard back from the three referees who accepted to evaluate the study. As you will see, the referees find the topic of your study of potential interest and are overall supportive. However, they raise a series of concerns and make suggestions for modifications, which we would ask you to carefully address in a revision of the present work.

In particular, reviewer #3 points out that the presented method should be discussed in the context of two important alternative technologies for bacterial genome engineering, namely temperature-sensitive plasmid integration and the CRISPR/Cas9 technology. Importantly, a major point raised by reviewers #2 and #3, refers to the need to carefully rewrite and streamline the manuscript in order to make the key findings and the major advantages offered by presented method, easily accessible to the readers. Careful editing is required so that additional findings that have been added in the revised version of the manuscript are closely integrated into the story.

 REFEREE REPORTS:

Reviewer #1 :

The authors experimentally confirmed the breadth and the depth of applications of this technique by adding more data. The text is more readable to general through the addition of more. The figures are generally more straightforward and easily-understood, except PCR as mentioned below.

Concerns:

1. It is still unclear how the structures of the lox sites affect the intron's efficiency.
2. Most of the PCR images are still not easily-understood for readers. For instance, the principle schematic of Figure 6A is only directly compatible with 6B but not 6C and 6D.

Final recommendation: accept after minor revisions.

Reviewer #2 :

The manuscript by Enyeart et al. is of high relevance for the community of bacteriologists who rely on easily manipulation of bacterial genomes to study the expression and functions of genes in their favorite bacterial species. The authors had already developed the "targetron" approach that is used by a number of bacteriologists to delete or replace genes in bacteria. In this manuscript, the strategy is exploited further to integrate the cre/lox system to the "targetron" approach and thus create a tool to make large insertions, deletions, and inversions in bacterial genomes with the advantage of not

including any antibiotic resistance cassette on the manipulated genome.

The authors have replied very carefully and in detail to all concerns of the reviewers.

Additional minor points:

- The first two points of the bullet points seem to be redundant.
- The authors should specify more clearly the need of constructing plasmids (sometimes up to three) for the use of the newly improved targetron system.
- The manuscript is very well written but lengthy in some parts. The reader could lose the focus of the strategies developed.

Reviewer #3 :

The authors present a novel and innovative combination of targetrons and recombinase-mediated cassette exchange for genome engineering in bacteria. In particular, the use of two different targetrons to deliver loxP sites in order to avoid excess homology is both elegant and parsimonious. The authors utilize Cre recombination to accomplish significant genome rearrangements, including large inversions and "cut-and-paste" that are not possible using alternative methods. The data demonstrating the effectiveness of their method is convincing. Perhaps most importantly, the flexibility and broad applicability of targetrons imply that this method can conceivably be extended to a wide variety of bacterial species that are currently amenable to few if any alternative genome engineering methods.

That said, two areas of this manuscript are especially concerning.

First is the authors' presentation of the significance of their method. While they are quite correct in emphasizing its broad applicability across many species and its markerless nature, the manuscript does not mention or cite two important alternative technologies for bacterial genome engineering that are critical to providing context for the reader.

Temperature-sensitive plasmid integration is a well-established method of gene deletion and replacement that has been utilized in a wide variety of species. The recent proliferation of improved plasmid cloning techniques such as isothermal/Gibson assembly has rendered temperature-sensitive vector construction with appropriate homology arms easier than ever. Of the organisms engineered in this manuscript, temperature-sensitive plasmids are available and have been used for this purpose in *E. coli* (Hamilton et al. 1989 *J. Bacter.* 171(9):4617-4622), *B. subtilis* (Biswas et al. 1993 *J. Bacter.* 175(11):3628-3635), and *S. aureus* (Luchansky et al 1989 *Mol. Microb.* 3(1):65-78), but are not currently available in *Shewanella*. Because this method is perhaps the most frequently utilized by microbiology laboratories working with organisms in which lambda Red recombineering does not function, it consequently represents the most appropriate yardstick for comparison. It will be important to note that many organisms such as *Shewanella* do not have temperature-sensitive plasmids available, and the generation of such a plasmid by targeted mutagenesis and screening is non-trivial for many laboratories. In these cases, most will likely find it easier to use the authors' targetron-and-Cre method.

With respect to markerless editing techniques, the advent of RNA-guided negative selection via Cas9 has in principle removed the need for markers in those methods that previously required them - most notably recombineering. It is true that Cas9-mediated negative selection is comparatively new (Jiang et al. 2013 *Nat. Biotech.* 31(3):233-239), but it is already widely used by many of the molecular, micro, and synthetic biology laboratories likely to consider adopting the authors' combined targetron-and-Cre approach. Cas9 also appears to function in essentially all microbes tested to date as well as in numerous eukaryotic organisms. Importantly, it has the potential to act synergistically with targetron insertion by selecting against cells with unmodified insertion sites, thereby increasing the effectiveness of low-activity targetrons in problematic organisms. This possibility might be mentioned in the discussion.

The second area of concern is the overall clarity of the manuscript. While the figures are appropriate and sufficiently informative (see minor suggestions noted below), reading the manuscript itself required significantly more effort than necessary. In general, too much information is included in the

main text; this most frequently occurs for the text colored red in the manuscript PDF. For example, the inclusion of an entire paragraph describing which factors influencing efficiency are statistically significant (p. 8) distracts the reader from simply looking at Figure 5B; it would be more appropriate to briefly describe the results and simply cite the Supplementary Text for those seeking more detailed information. Phrases such as "intron-mediated insertions of lox sites" could be streamlined to "intron-delivered lox sites", while "genome editing" as the name of a method is far too general; almost every genome engineering method refers to "editing" the genome. That the non-E.coli work was added at a later date is a bit too obvious (or if it was not, it reads as though it were) and could be integrated more closely into the overall story. Overall, the manuscript text could use some careful editing and streamlining throughout its length, with an emphasis on the key advantages offered by the authors' new method: inversions, cut-and-paste, and functionality in organisms that are not amenable to recombineering or temperature-sensitive plasmid-based techniques.

Given the extensive evidence of targetron and Cre functionality across diverse microbes, there is no need for further experiments to more extensively document these advantages. However, a discussion of the circumstances in which large-scale inversions and cut-and-paste will be useful to other laboratories would be highly instructive, while a frank appraisal of how many microbes are amenable to targetron-and-Cre but not recombineering or temperature-sensitive plasmids would do much to clarify the potential impact of this study. Combined with a carefully edited and streamlined manuscript text, these elements would greatly increase the suitability of publication.

Suggested figure changes:

Figures 6-7: Careful reading of the figure legends is required to understand 6c, 6d, 7c, and 7d. The figures would be clearer if a small and simplified version of the U strain diagram depicted in 6a/7a was included just above each gel image.

Figure 7a: In the "U" strain diagram, 5' and 3' would ideally all be written in the normal orientation rather than inverted.

2nd Revision - authors' response

20 July 2013

We have made a number of improvements in this draft, including significantly streamlining the results sections, improving the clarity of **Figures 6 and 7**, and adding more analysis of applications and the utility of the method relative to suicide plasmids and the CRISPR-Cas9 system to the discussion. We have also given the method a more specific name, Genome Editing via Targetrons and Recombinases (GETR). The changes we have made to the text relative to the last submitted version are highlighted in red (when only the first letter of a sentence is highlighted, that means text previously positioned before that sentence has been deleted in the new draft).

Reviewer #1

It is still unclear how the structures of the lox sites affect the intron's efficiency.

In an effort to further clarify this issue, we have added the following text to p. 6, lines 15-17:

"We hypothesize that this trend occurs because inserts having inflexible structures are more likely to interfere with proper folding of the catalytic structures of the intron than are inserts having flexible structures, which can be moved away from other formations."

Structural studies would be required to more fully explore this hypothesis, but such studies are beyond the scope of the present work.

Most of the PCR images are still not easily-understood for readers. For instance, the principle schematic of Figure 6A is only directly compatible with 6B but not 6C and 6D.

We have added small schematics to **Figures 6C, 6D, 7C, and 7D** to clarify exactly what PCR products are being assessed, and we hope this will resolve the issue.

Reviewer #2***The first two points of the bullet points seem to be redundant.***

We have changed the text of the second bullet-point to "Targetrons and Cre/lox together represent a broad-host range solution to genome editing" to better emphasize that point and differentiate it from the first point.

The authors should specify more clearly the need of constructing plasmids (sometimes up to three) for the use of the newly improved targetron system.

We have modified the sentence on p. 23, lines 23 to p. 24, line 3, to read: "Though we used plasmids (one plasmid carrying the targetron, one plasmid carrying the Cre gene, and, as necessary, a plasmid or other vector carrying DNA to be integrated, delivered by electroporation or conjugation) to deliver targetrons in the present study, phage, direct electroporation or other methods could potentially be used, as well."

The manuscript is very well written but lengthy in some parts. The reader could lose the focus of the strategies developed.

We have removed more of the statistical discussion to the supplementary materials, and have significantly scaled back the discussion of **Figures 6** through **8** in the sections entitled "Deletions," "Inversions," and "One-Step Cut-and-Paste" on pages 10 through 15. We hope this results in a much more readable manuscript.

Reviewer #3

Temperature-sensitive plasmid integration is a well-established method of gene deletion and replacement that has been utilized in a wide variety of species. The recent proliferation of improved plasmid cloning techniques such as isothermal/Gibson assembly has rendered temperature-sensitive vector construction with appropriate homology arms easier than ever. Of the organisms engineered in this manuscript, temperature-sensitive plasmids are available and have been used for this purpose in *E. coli* (Hamilton et al. 1989 *J. Bacter.* 171(9):4617-4622), *B. subtilis* (Biswas et al. 1993 *J. Bacter.* 175(11):3628-3635), and *S. aureus* (Luchansky et al 1989 *Mol. Microb.* 3(1):65-78), but are not currently available in *Shewanella*. Because this method is perhaps the most frequently utilized by microbiology laboratories working with organisms in which lambda Red recombineering does not function, it consequently represents the most appropriate yardstick for comparison. It will be important to note that many organisms such as *Shewanella* do not have temperature-sensitive plasmids available, and the generation of such a plasmid by targeted mutagenesis and screening is non-trivial for many laboratories. In these cases, most will likely find it easier to use the authors' targetron-and-Cre method.

We have added the following text to the Discussion, page 20, lines 3-15:

"Another common method of genome engineering is the use of suicide plasmids. For instance, temperature-sensitive integrable plasmids have been developed for all the systems described here other than *S. oneidensis* (Biswas et al, 1993; Hamilton et al, 1989; Link et al, 1997; Luchansky et al, 1989), and systems based on plasmids requiring expression of the *pir* (Kolter et al, 1978; Miller & Mekalanos, 1988) or *repA* (Leenhouts et al, 1996) genes to replicate have also been frequently employed. These systems are most useful for gene replacements. For the types of modifications discussed in the present work, suicide plasmids present many of the same limitations as recombineering, such as requirements for selectable markers and large regions of homology, and are limited by poor efficiency and relatively high background. The profusion of research into alternative recombineering systems in recent years, described above, is symptomatic of broader dissatisfaction with suicide plasmids as genetic tools, and the present system represents a favorable alternative for large-scale genomic modifications."

Additionally, as mentioned on p. 4, lines 19-21, the insufficiency of suicide plasmids as genomic modification tools has been a primary motivating factor in the development of targetrons for many systems. For example, a Google Scholar search for mentions of "ClosTron" (the name for one of the targetron mutagenesis systems developed for *Clostridium*) turns up 38 papers published in 2013 to

date (http://scholar.google.com/scholar?q=clostron&as_ylo=2013&as_yhi=2013), indicating that the community has found the targetron approach to be of great utility as compared to previous tools. We do not, however, consider the lack of temperature-sensitive plasmids for the *Shewanella* system to be a limiting factor for work in this strain, as other replication-deficient plasmids, such those based on the *pir* system are routinely introduced via conjugation (see for instance Bouhenni et al. Appl. Environ. Microbiol. 71(8): 4935–4937 (2005)), and this approach can also be used in many other systems for which temperature-sensitive plasmids have not been developed. We do not consider this fact as detrimental to the utility of our system, which we consider to be a favorable alternative to suicide plasmids for nearly all genomic modifications other than allele replacement.

With respect to markerless editing techniques, the advent of RNA-guided negative selection via Cas9 has in principle removed the need for markers in those methods that previously required them - most notably recombineering. It is true that Cas9-mediated negative selection is comparatively new (Jiang et al. 2013 Nat. Biotech. 31(3):233-239), but it is already widely used by many of the molecular, micro, and synthetic biology laboratories likely to consider adopting the authors' combined targetron-and-Cre approach. Cas9 also appears to function in essentially all microbes tested to date as well as in numerous eukaryotic organisms. Importantly, it has the potential to act synergistically with targetron insertion by selecting against cells with unmodified insertion sites, thereby increasing the effectiveness of low-activity targetrons in problematic organisms. This possibility might be mentioned in the discussion.

We have added the following text to the Discussion, p. 20, line 16, to p. 21, line 17:

"A more recent addition to the set of tools available for genome engineering is the CRISPR/Cas9 system, which adapts the site-specific RNA-mediated restriction system of bacteria toward making targeted double-strand breaks in genomic DNA (Cho et al, 2013; Cong et al, 2013; Hwang et al, 2013; Mali et al, 2013). Methods of genome engineering relying solely on the creation of double-strand breaks have not traditionally gained much traction in bacterial systems. Besides the requirement for selectable markers, the efficiency of double-strand break repair tends to be poor in bacteria, since most prokaryotes are only capable of repairing breaks via homologous recombination, and those that can carry out non-homologous end joining have only a rudimentary system for doing so (Aravind & Koonin, 2001; Hefferin & Tomkinson, 2005). CRISPR-Cas9-mediated cutting of genomic DNA has been shown to be lethal to bacteria (Bikard et al, 2012), but Jiang and coworkers have recently reported that this method can be used to select for the integration of mutated DNA homologous to the cut site (Jiang et al, 2013).

However, the CRISPR/Cas9 system alone is only of functional efficiency in bacteria that have very active recombination systems, such as *Streptococcus pneumoniae*, and in those systems the CRISPR-Cas9 expression construct must also be integrated into the genome along with a selectable marker and then subsequently removed. In *E. coli* the CRISPR-Cas9 system has been shown to increase the efficiency of recombineering by cleaving the genome at unmodified sites (and thereby selecting for modified strains), but this method also has the inherent limitations of recombineering; i.e., the requirement of a mutator strain for high efficiency, limitation to relatively small changes, and generally poor efficiency in systems other than *E. coli*. That said, it is possible that a more general application of CRISPR-Cas9 could be to increase the efficiency targetron-mediated mutagenesis. Finally, recent work by Fu and coworkers (Fu et al, 2013) demonstrates extensive off-target mutagenesis by CRISPR-Cas9, often at efficiencies comparable to the degree of on-target mutagenesis."

In general, too much information is included in the main text; this most frequently occurs for the text colored red in the manuscript PDF. For example, the inclusion of an entire paragraph describing which factors influencing efficiency are statistically significant (p. 8) distracts the reader from simply looking at Figure 5B; it would be more appropriate to briefly describe the results and simply cite the Supplementary Text for those seeking more detailed information.

We have removed more of the statistical discussion to the supplementary materials, and have significantly scaled back the discussion of **Figures 6** through **8** in the sections entitled "Deletions," "Inversions," and "One-Step Cut-and-Paste" on pages 10 through 15. We believe this results in a much more readable manuscript.

Phrases such as "intron-mediated insertions of lox sites" could be streamlined to "intron-delivered lox sites"

We have done our best to amend such instances, and rely on the editors to aid in eliminating similar problems that we may have missed.

"Genome editing" as the name of a method is far too general; almost every genome engineering method refers to "editing" the genome.

We concede this point, and have renamed the method "Genome Editing via Targetrons and Recombinases (GETR)."

That the non-E.coli work was added at a later date is a bit too obvious (or if it was not, it reads as though it were) and could be integrated more closely into the overall story.

We are not entirely sure what the Reviewer is referring to. We specify the utility of Targetrons for non-*E. coli* species in both the Introduction and the Discussion. Like many researchers, we developed proofs-of-principle for *E. coli* in advance of applying the work to other, less tractable species. We would suggest that that lack of integration between species is actually just a logical progression from proofing the system to applying it.

A discussion of the circumstances in which large-scale inversions and cut-and-paste will be useful to other laboratories would be highly instructive.

Concerning inversions, we have added the following text to the discussion on p. 22, lines 4-9:

"While large-scale inversions were presented here primarily as a demonstration of the lack of size limits for generating rearrangements using our method, artificial inversions have traditionally been used for studying genome structure and its constraints (Campo et al, 2004; Esnault et al, 2007; Garcia-Russell et al, 2004; Guijo et al, 2001; Hill & Gray, 1988; Rebollo et al, 1988; Segall et al, 1988; Valens et al, 2004), and the approaches presented herein allow such studies to be more easily performed in many more systems."

Concerning one-step cut-and-paste, we have moved the paragraph concerning this technique to directly after the discussion of inversions on p. 22, lines 10-21, and modified the paragraph to provide more explicit details on potential applications:

"The one-step cut-and-paste method we present is of particular interest given that it allows one piece of a genome to be inserted within another site, without the accumulation of intervening intermediates, an operation that is not possible with any other technique. The cut-and-paste method could also be applied to more nuanced studies of genome structure constraints. For instance, the effect of moving different structural domains or of swapping two domains, such as the Ori and Ter domains, could be examined. Additionally, expression levels tend to be dependent on genomic location, with, for instance, genes nearer the origin tending to be more highly expressed (Cooper & Helmstetter, 1968; Rocha, 2008), and thus cut-and-pastes could be used as a simple means for modulating the overall expression levels of super-operons (Lathe et al, 2000; Rogozin et al, 2002) or other large genetic units. The ability to move DNA between species without regard for inherent similarities or phylogenetic relationships opens up the possibility of using genomic editing for rapidly adapting bacterial genomes."

A frank appraisal of how many microbes are amenable to targetron-and-Cre but not recombinering or temperature-sensitive plasmids would do much to clarify the potential impact of this study.

As mentioned earlier, we believe the variety of conditionally-replicating plasmids available provide suitable alternatives to temperature sensitive plasmids, but except for gene replacements we do not consider these to be significant competitors with our work.

Figures 6-7: Careful reading of the figure legends is required to understand 6c, 6d, 7c, and 7d. The figures would be clearer if a small and simplified version of the U strain diagram depicted in 6a/7a was included just above each gel image.

We have complied with this request and agree and it improves the understandability of these figures.

Figure 7a: In the "U" strain diagram, 5' and 3' would ideally all be written in the normal orientation rather than inverted.

We have done so.